# Crystal symmetry modification enables high-ranged in-plane thermoelectric performance in n-type SnSe crystals

Haonan Shi[1,2,3,9], Yi Wen[1,9], Shulin Bai[1], Cheng Chang[1], Lizhong Su[4], Tian Gao[1], Shibo Liu[1], Dongrui Liu[1], Bingchao Qin[1], Yongxin Qin[1], Huiqiang Liang[5], Xin Qian[5], Zhenghao Hou[6], Xiang Gao[7], Tianhang Zhou[8] ✉, Qing Tan[1] ✉ & Li-Dong Zhao[1,2,3] ✉

SnSe crystal has witnessed significant advancements as a promising thermoelectric material over the past decade. Its in-plane direction shows robust mechanical strength for practical thermoelectric applications. Herein, we optimize the in-plane thermoelectric performance of *n*-type SnSe by crystal symmetry modification. In particular, we find that Te and Mo alloying continuously enhances the crystal symmetry, thereby increasing the carrier mobility to ~ 422 $cm^2 V^{-1} s^{-1}$. Simultaneously, the conduction bands converge with the symmetry modification, further improving the electrical transport. Additionally, the lattice thermal conductivity is limited to ~ 1.1 $W m^{-1} K^{-1}$ due to the softness of both acoustic and optical branches. Consequently, we achieve a power factor of ~ 28 $\mu W cm^{-1} K^{-2}$ and *ZT* of ~ 0.6 in *n*-type SnSe at 300 K. The average *ZT* reaches ~ 0.89 at 300−723 K. The single-leg device based on the obtained *n*-type SnSe shows a remarkable efficiency of ~ 5.3% under the $\Delta T$ of ~ 300 K, which is the highest reported in *n*-type SnSe. This work demonstrates the substantial potential of SnSe for practical applications of power generation and waste heat recovery.

The escalating demand for energy and the challenges posed by the environment necessitate the exploitation of innovative energy sources. Researchers have focused on achieving ordered energy conversion in recent decades, exemplified by renewable electrified heating[1], powered decarbonized technology[2], etc. As a representative example, thermoelectric technology enables effective management and reversible conversion of electrical and thermal energy[3–5]. It is crucial to optimize electrical transport while minimizing heat transport for thermoelectric materials, as indicated by the thermoelectric figure of merit $ZT = \sigma S^2 T / \kappa$, where $\sigma$, $S$, $T$, and $\kappa$ are electrical conductivity, Seebeck coefficient, temperature, and thermal conductivity[6,7].

The above parameters are strongly coupled, posing a significant challenge in their simultaneous optimization[8]. Effective strategies have been proposed to manage the complex interrelations among thermoelectric parameters, including carrier concentration tuning[9–11], band structure engineering[12,13], all-scale hierarchical defects[14,15], etc.

The thermoelectric properties strongly depend on the crystal structure, therefore, the structure modification can be applied to

[1]School of Materials Science and Engineering, Beihang University, Beijing, China. [2]Center for Bioinspired Science and Technology, Hangzhou International Innovation Institute, Beihang University, Hangzhou, China. [3]Tianmushan Laboratory, Beihang University, Hangzhou, China. [4]School of Materials Science and Engineering, Taiyuan University of Science and Technology, Taiyuan, China. [5]College of Physical Science and Technology, Hebei University, Baoding, China. [6]Shijiazhuang Key Laboratory of Low Carbon Energy Materials, College of Chemical Engineering, Shijiazhuang University, Shijiazhuang, China. [7]Center for High-Pressure Science and Technology Advanced Research (HPSTAR), Beijing, China. [8]College of Carbon Neutrality Future Technology, State Key Laboratory of Heavy Oil Processing, China University of Petroleum (Beijing), Beijing, China. [9]These authors contributed equally: Haonan Shi, Yi Wen. ✉e-mail: zhouth@cup.edu.cn; tanqing@ustb.edu.cn; zhaolidong@buaa.edu.cn

compromise the inherent contradiction between electrical and thermal transports[16]. Symmetry, as a crucial factor of the crystal structure, has been extensively explored in thermoelectrics[17]. Both symmetry-enhancing[15] and symmetry-breaking[18] have been successfully implemented to improve thermoelectric properties in various material systems. On one hand, high-symmetry materials typically exhibit favorable electrical transport because of their high valley degeneracy and carrier mobility; however, they tend to come with the drawback of higher thermal conductivity[19]. Researchers have reduced the thermal conductivity by disrupting their local symmetry without impeding the carrier transport[20]. For instance, Ge-alloyed PbSe forms off-centered lattice points, leading to soft phonon modes and strongly suppressed thermal conductivity[21]. On the other hand, several promising thermoelectric materials possess low symmetry structures resulting in intrinsically low thermal conductivity[22]. For such materials, enhancing the structure symmetry to boost carrier mobility or promote electronic band convergence has been confirmed to be an effective strategy[23,24]. For example, increasing the entropy of GeTe enhances its symmetry, resulting in the converged bands[15]. Similar approaches have been successfully applied to other thermoelectric materials such as GeSe[25], BaAgSb[26], and $Cu_2SnSe_3$[27].

SnSe, as a typical low-symmetry material, has experienced extensive investigations into its symmetry in recent years[23,28,29]. Previous studies have revealed that a continuous phase transition occurs in SnSe from ~600 K to ~800 K[30], during which its symmetry progressively increases. Meanwhile, converging and diverging behaviors also happen in both valence and conduction bands with symmetry changing[30,31]. Therefore, crystal symmetry modification is recognized as a crucial strategy to improve the thermoelectric properties of SnSe. It has been well-established that Pb can effectively improve the symmetry and elevate the phase transition temperature of SnSe, leading to excellent electrical transport properties[30,32,33]. Te-alloying has also been implemented in p-type SnSe, yielding a more symmetric structure[23]. The incorporation of high-symmetry materials into SnSe in large proportions has been explored to obtain cubic SnSe, such as $AgSb(Te/Se)_2$[34–36], $NaSbTe/Se_2$[37], and $AgBiTe/Se_2$[38,39]. Furthermore, the nonequilibrium process has been considered to generate highly symmetric SnSe[40,41]. Theoretical calculations have also contributed significantly to advancing the symmetry study of SnSe[42,43].

In this work, we focused on investigating the correlation between symmetry and thermoelectric properties by stepwise alloying Te and Mo into n-type SnSe crystals. Our emphasis was evaluating the in-plane thermoelectric performance, as illustrated in Supplementary Fig. S1b. Although n-type SnSe crystals have performed excellent thermoelectric performance along the out-of-plane direction, the cleavaged characteristic makes it difficult for application. In contrast, the in-plane direction has advantages in mechanical strength, weldability, and electroplatability, which is more suitable for device manufacturing[44]. It is worth noting that all the samples were doped with 3% Br to realize the n-type electronic transport in this work. To acquire accurate crystal structure data, synchrotron radiation was employed, revealing a continuous improvement in the symmetry of SnSe with the addition of Te and Mo. We observed that the convergence of two conduction bands occurs during Te alloying, benefiting from the interaction between crystal structure and electronic energy band behavior in SnSe (Fig. 1a). Resultantly, the carrier mobility is boosted to ~422 $cm^2V^{-1}s^{-1}$ with the addition of Te and Mo, maintaining the degenerated conduction bands. The power factor (PF) reaches a notable ~28 $\mu W\,cm^{-1}\,K^{-2}$ at 300 K, indicating a prominent improvement in electrical transport property. Meanwhile, the analysis of phonon dispersion shows limited contributions in both acoustic and optical branches, resulting in a reduced lattice thermal conductivity of ~1.1 $W\,m^{-1}\,K^{-1}$ at room temperature. The electron-phonon transport is well decoupled through the combined effects of Te and Mo (Fig. 1b). Finally, a saturated ZT curve is obtained in SnSe-0.75%Te-0.7%Mo, which possesses a room-temperature ZT of ~0.6 and an average ZT of ~0.89 at 300−723 K (Fig. 1c). Furthermore, a single-leg maximum conversion efficiency (η) of ~5.3% (ΔT ~ 300 K) is obtained in the optimized composition

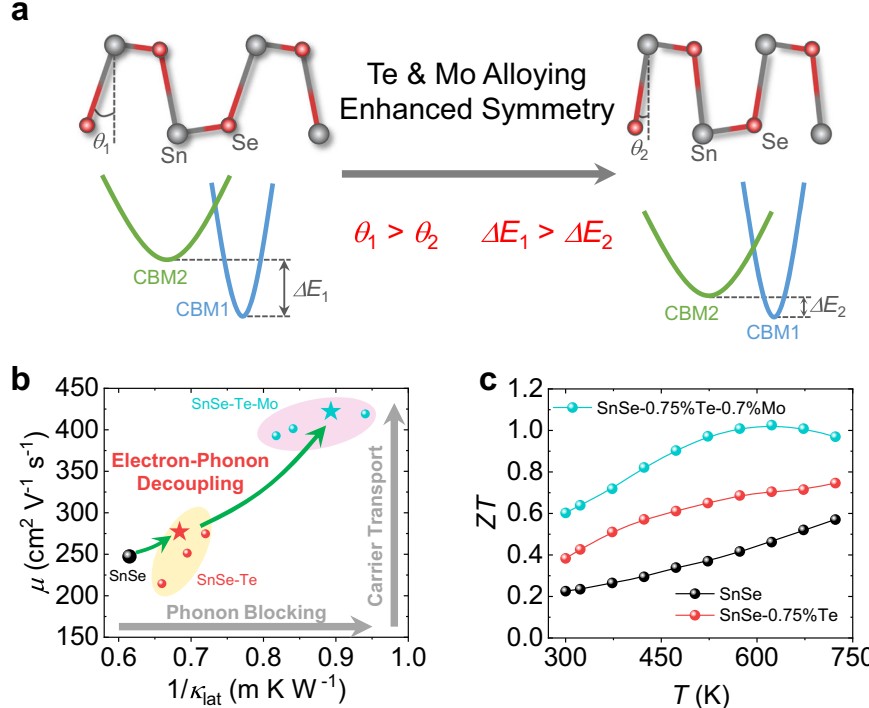

**Fig. 1 | High-performance n-type SnSe crystals with Te and Mo alloying.**
**a** Schematic illustration of symmetry enhancement by Te and Mo alloying, accompanied by the band convergence. **b** The relationship between carrier mobility and the inverse of lattice thermal conductivity in SnSe-Te and SnSe-Te-Mo. **c** ZT of SnSe, SnSe-0.75%Te, and SnSe-0.75%Te-0.7%Mo at 300−723 K.

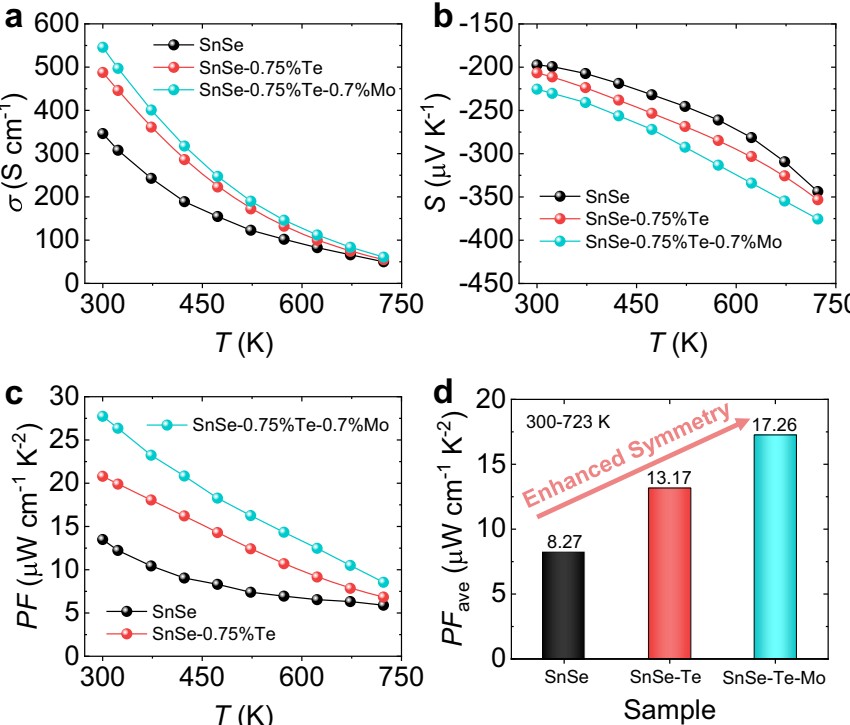

**Fig. 2 | The electrical transport properties of SnSe, SnSe-0.75%Te, and SnSe-0.75%Te-0.7%Mo at 300 − 723 K. a** Electrical conductivity. **b** Seebeck coefficient. **c** Power factor (*PF*). **d** The average power factor (*PF*_ave).

$Sn_{0.993}Mo_{0.007}Se_{0.9625}Te_{0.0075}Br_{0.03}$. This work underscores the feasibility of modifying the crystal symmetry in SnSe crystals, which holds promise for generalization to other thermoelectric materials, particularly those with low crystal symmetry. The high-performance *n*-type SnSe crystals in this work are expected to give impetus for the application of SnSe crystals and contribute significantly to the development of renewable electricity.

## Results
### Electrical transport properties
Herein, the Bridgman method was employed to synthesize two series of *n*-type SnSe crystal samples, namely $SnSe_{0.97-x}Te_xBr_{0.03}$ ($x = 0$, 0.5%, 0.75%, 1%, 1.25%) and $Sn_{1-y}Mo_ySe_{0.9625}Te_{0.0075}Br_{0.03}$ ($y = 0$, 0.3%, 0.5%, 0.7%, 0.9%), referred to as SnSe-*x*Te and SnSe-0.75%Te-*y*Mo respectively for simplicity in this work. The photograph of the actual object of the obtained crystal and a schematic illustrating the in-plane direction are presented in Supplementary Fig S1. X-ray diffraction (XRD) was conducted to identify the phase of samples, which were found to exhibit a single phase of *Pnma*, as presented in Supplementary Fig. S2, and the lattice parameters exhibit a near-linear variation. The EDS mapping was also conducted to confirm that the element distribution is uniform in samples as shown in Supplementary Fig S6. Results of XRD and EDS mapping collectively confirm that Te and Mo have been successfully alloyed into the matrix.

The electrical transport properties of SnSe, SnSe-0.75%Te, and SnSe-0.75%Te-0.7%Mo are shown in Fig. 2, which are the two most optimized samples. The detailed information on electrical transport properties is shown in Supplementary Figs. S3a−c and S4a−c. As a prerequisite for the analysis, the metal-like electrical conductivities (σ) shown in Fig. 2a and the negative Seebeck coefficient (*S*) depicted in Fig. 2b confirm the heavy-doped *n*-type SnSe. In Fig. 2a, σ exhibits a significant increase across the entire temperature range with a particularly pronounced rise from ~346 S cm⁻¹ in pure SnSe to ~487 S cm⁻¹ in SnSe-0.75%Te, and further to ~545 S cm⁻¹ in SnSe-0.75%Te-0.7%Mo at

room temperature. Surprisingly, Fig. 2b demonstrates an increase of *S* after Te and Mo alloying. The synergetic improvement of σ and *S* leads to a boosted *PF* in the wide temperature range, peculiarly around near-room temperature (~28 μW cm⁻¹ K⁻² at 300 K), as shown in Fig. 2c. Moreover, compared to pure SnSe crystals, the average *PF* value at 300−723 K is doubled (~17.26 μW cm⁻¹ K⁻²) in SnSe-0.75%Te-0.7%Mo as shown in Fig. 2d, indicating that both Te and Mo contribute collectively towards advancing the electrical transport properties of *n*-type SnSe crystals.

To further understand the change in electrical transport, the carrier concentration (*n*) and mobility (*μ*) of SnSe-*x*Te and SnSe-0.75% Te-*y*Mo were measured as shown in Fig. 3. The initial *n* of ~8.75 × 10¹⁸ cm⁻³ in SnSe is increased to over 1 × 10¹⁹ cm⁻³ in SnSe-*x*Te while maintaining a high in-plane *μ* in Fig. 3a. Subsequently, we analyzed the change of the bandgap (*E*_g) using the ultraviolet diffuse reflection spectrum depicted in Fig. 3b. The leftward shift of the intercept on the horizontal axis demonstrates a reduced *E*_g value in SnSe-*x*Te, leading to a shallower defect level[9]. As a result, more electrons are motivated to the conduction band to participate in the electrical transport, thus increasing the *n*.

In Fig. 3c, the *n* shows a decreasing trend and stabilizes at around ~8 × 10¹⁸ cm⁻³ with Mo addition, and the *μ* is significantly enhanced from ~277 cm²V⁻¹ s⁻¹ in SnSe-0.75%Te to ~422 cm²V⁻¹ s⁻¹ in SnSe-0.75%Te-0.7%Mo. Given that Mo is a multivalent element, it is crucial to specify its valance state in SnSe-Te-Mo. X-ray photoelectron spectroscopy (XPS) was employed for this purpose and the spectra are shown in Fig. 3d and Supplementary Fig. S7. The binding energy of C 1*s* is 284.8 eV for calibration in Supplementary Fig. S7b. The observed peak of Mo $3d_{5/2}$ at 228.9 eV indicates the existence of Mo²⁺ in SnSe-Te-Mo[45,46]. However, Mo²⁺ and Sn²⁺, although being isoelectronic, have different electronegativity (χ), with $\chi_{Mo}$ (2.16) being larger than $\chi_{Sn}$ (1.96)[47]. That is, Mo is more prone to attract electrons than Sn, resulting in fewer free electrons in the matrix and accounting for the decreased carrier concentration as shown in Fig. 3c.

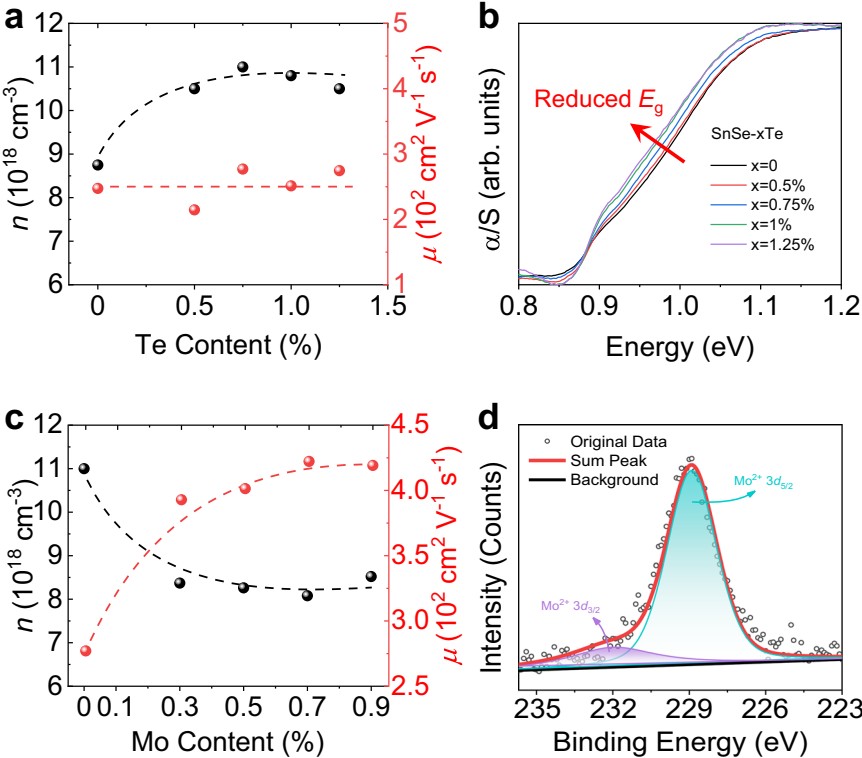

**Fig. 3 | The variation of carrier concentration and carrier mobility of SnSe-$x$Te and SnSe-0.75%Te-$y$Mo. a** Carrier concentration and carrier mobility as a function of Te content. **b** Ultraviolet diffuse reflection spectrum. **c** Carrier concentration and carrier mobility as a function of Mo content. **d** X-ray photoelectron spectroscopy (XPS) spectra of Mo element in SnSe-0.75%Te-0.7%Mo.

## High symmetry accompanied by band convergence

As previously discussed, the $\sigma$ and $S$, which typically exhibit competing behaviors, are simultaneously enhanced upon Te and Mo alloying. In-depth analyses were conducted to further elucidate the underlying optimization mechanism. We employed the Pisarenko relationship as depicted in Fig. 4a. It can be observed that the density of state effective mass ($m^*$) increases from ~0.42$m_e$ in SnSe under single band conduction to ~0.51$m_e$ upon Te alloying. This value is consistent with our previous work[48], indicating the involvement of two bands for conduction after Te alloying. To further describe the situation of the energy bands, SR-XRD was carried out to examine the detailed structure of SnSe and SnSe-0.75%Te samples from 300 to 723 K. The SR-XRD patterns containing detailed structure information are shown in Supplementary Figs. S10 and S11, and the resulting electronic band structures are shown in Supplementary Figs. S13 and S14 by the density functional theory (DFT) calculations. Fig. 4b presents a comparison between the band structures of SnSe and SnSe-0.75%Te at room temperature, and the energy difference ($\Delta E$) of the two conduction bands decreases from ~0.13 eV in SnSe to ~0.1 eV in SnSe-0.75%Te. A reduction in $\Delta E$ by more than 20% demonstrates the degeneracy of the conduction bands. The valley degeneracy ($N_V$) increases with the second conduction band participating in the electrical transport, therefore enhancing the $m^*$, as $m^* = N_V^{2/3} m_b^*$, where $m_b^*$ is the single-band effective mass. It needs to be pointed out that the band convergence provides more carrier transport channels, however, increases the intervalley scattering[49,50]. Thus there is no significant increase in carrier mobility as discussed in Fig. 3a. The variation of $\Delta E$ with different temperatures was summarized in Fig. 4c. The $\Delta E$ decreases to different degrees throughout the entire temperature range with Te alloying rather than solely at room temperature, leading to the improved electrical transport properties from 300 K to 723 K. The two conduction bands keep converging after Mo adding, as shown in Supplementary Fig. S15. Therefore, the increased $S$ in SnSe-Te is mainly

contributed to the enhanced $m^*$ lead by the band convergence. The behavior of the energy band can be attributed to the change in the crystal structure, and we will discuss below.

The introduction of Mo caused a dramatical increase in $\mu$ as shown in Fig. 4d. To get an insight into the origin of the increased $\mu$, we analyzed the lattice parameters at different temperatures in SnSe, SnSe-0.75%Te, and SnSe-0.75%Te-0.7%Mo in Fig. 4e, which was derived from the refinement of SR-XRD data. The relative information is provided in Supplementary Figs. S10–S12. The detailed structural data of the three samples is also provided in Supplementary Tables S2–S4. The lattice expands along the $a$ and $b$ directions with Te and Mo addition, while contracts along the $c$ direction. This variation trend is also consistent with the data in Supplementary Figs. S2c, d. This suggests that the symmetry of the lattice is enhanced[30]. More vividly, we describe the change in the bond angle in Fig. 4f. The angle between the $a$ direction and the Sn-Se bond is defined as $\theta$, shown in the inset of Fig. 4f. A smaller $\theta$ indicates a more symmetric crystal structure, and when the $\theta$ decreases to zero, the structure turns into *Cmcm*[51]. Results revealed that the $\theta$ decreases slightly in SnSe-0.75%Te compared to SnSe and drops sharply in SnSe-0.75%Te-0.7%Mo. The reduction in $\theta$ may be attributed to the differences in ion radii between extrinsic ions (Mo$^{2+}$ and Te$^{2-}$) and host ions (Sn$^{2+}$ and Se$^{2-}$), in which the radii of Sn$^{2+}$, Mo$^{2+}$, Se$^{2-}$, and Te$^{2-}$ are 118 pm, 73 pm, 198 pm, and 221 pm, respectively[52]. The addition of Te and Mo strengthens the strain in the crystal and forces the host Sn and Se to move closer along the $c$ direction, which fits in with the trend of $c$ observed in Fig. 4e, ultimately leading to a smaller $\theta$ and higher symmetry[28]. Therefore, Mo alloying significantly boosts the $\mu$ by enhancing the structure symmetry in $n$-type SnSe crystals, resulting in further improved electrical transport properties. In addition, we also considered the variation in out-of-plane mobility and found that it exhibits a similar trend as that along the in-plane direction as shown in Supplementary Table S5, demonstrating that symmetry enhancing can promote electrical transport properties

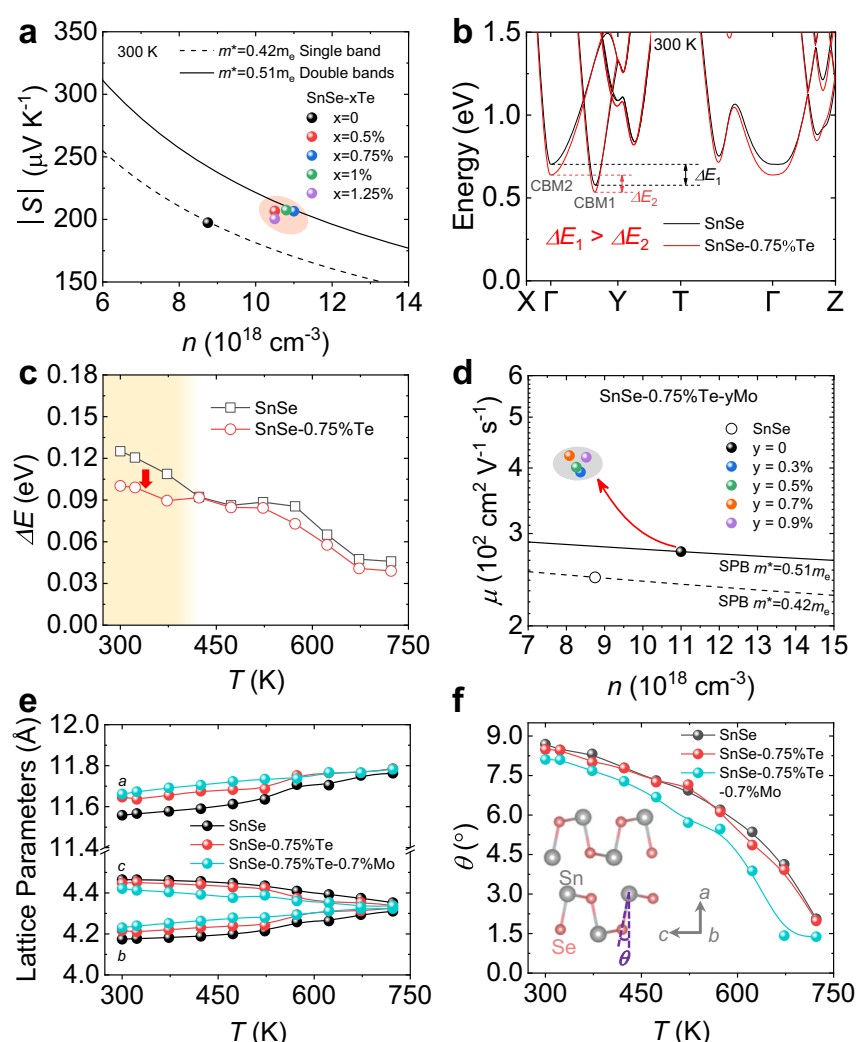

**Fig. 4 | Symmetry enhancement accompanied by the band convergence.**
**a** Seebeck coefficient as a function of carrier concentration. **b** The electronic band structure of SnSe and SnSe-0.75%Te at 300 K. **c** The energy difference of the two conduction bands ($\Delta E$) as a function of temperatures in SnSe and SnSe-0.75%Te. **d** Carrier mobility as a function of carrier concentration. **e** Lattice parameters as a function of temperatures in SnSe, SnSe-0.75%Te, and SnSe-0.75%Te-0.7%Mo. **f** The $\theta$ as a function of temperatures in SnSe, SnSe-0.75%Te, and SnSe-0.75%Te-0.7%Mo, while the $\theta$ is the angle between the Sn-Se bond and the $a$ direction, as shown in the inset.

along both directions, rather than being specific to any particular direction.

We can conclude that the co-alloying of Te and Mo facilitated the highly symmetric structure of $n$-type SnSe crystals. Simultaneously, the convergence of conduction bands when Te adding can be regarded as inevitable in the process of symmetry enhancing[29,30]. The weighted mobility ($\mu_W$) of SnSe-$x$Te and SnSe-0.75%Te-$y$Mo shows the upward trend in Supplementary Fig. S8, further confirming the synergetic optimization of electrical transport parameters. The higher symmetric structure accompanied by the converged bands realizes the decoupling of electrical transport parameters and high $PF$ in SnSe-0.75%Te-0.7%Mo.

**Microstructure characterization**

To further confirm the symmetry enhancement of SnSe, we observed the microstructure of SnSe-0.75%Te-0.7%Mo. A detailed aberration-corrected scanning transmission electron microscopy (AC-STEM) characterization is summarized in Fig. 5. For comparison, an annular dark field- (ADF-) STEM image of pristine SnSe viewed along [010] direction is given in Fig. 5a, showing a distorted rocksalt structure. Some regions with darker contrasts are most likely due to intrinsic Sn

vacancies. After Te&Mo co-alloying, an annular bright field- (ABF-) STEM image of such sample highlights the regions with darker contrast induced by strain (Fig. 5b). Enlarged ADF-STEM images of a representative strained region (Fig. 5c, d) expose a distorted region with darker contrast within the lattice. An intensity profile across Fig. 5d shows a decrease in ADF-STEM image intensity at the Sn sites within the distorted region (Fig. 5e). Since Sn vacancy will not induce such lattice distortion, the decreased intensity is most likely due to Mo substitution at Sn sites. We selected a highly distorted region in Fig. 5c and compared it with a reference region in Fig. 5a, the difference in lattice symmetry is obvious, which results in a discernible difference in $\theta$ as shown in Fig. 5f1, 2. For the reference region, $\theta \sim 8.63°$, close to the value in Fig. 4f, and for the distorted region, $\theta \sim 2.30°$, which is significantly decreased and indicates the symmetry enhancement. To note, the distorted $\theta$ here is not the same as that in Fig. 4f, because the $\theta$ here is observed in the specific region while the $\theta$ obtained by refinement reflects the average value in the lattice. However, the decreased $\theta$ as a sign of the high symmetry can be confirmed by the refinement and microstructure observation.

Using inverse fast Fourier transform (IFFT) of selected reflections from Fig. 5d, we obtained IFFT images reconstructed using signals

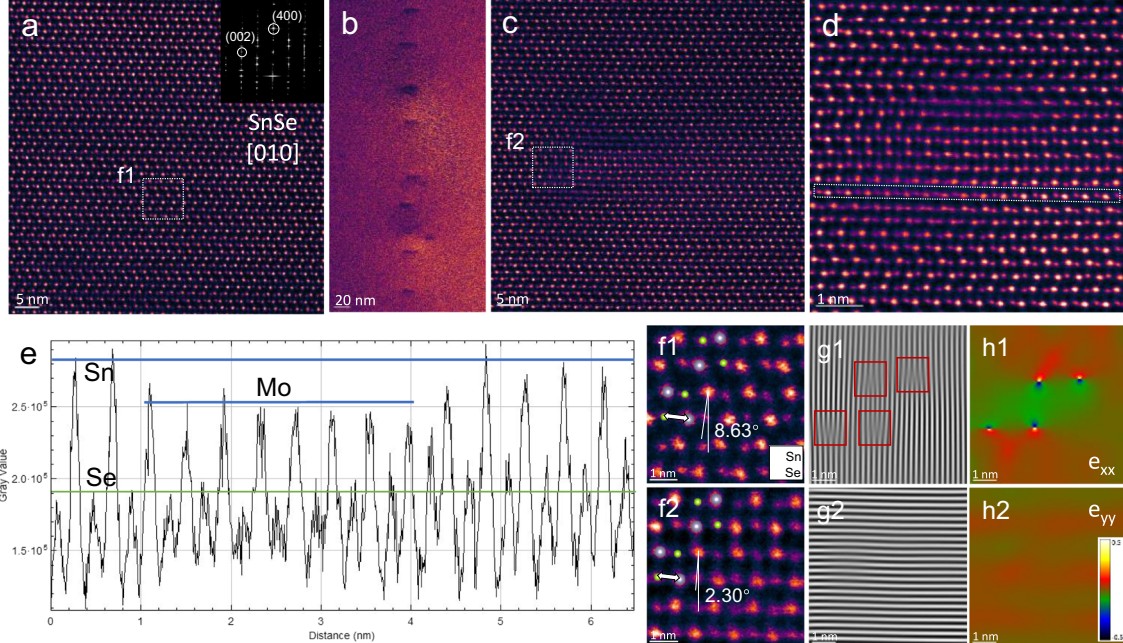

**Fig. 5 | AC-STEM characterization of SnSe before and after Te/Mo co-doping.**
**a** ADF-STEM image of pristine SnSe viewed along [010] direction, inset showing the corresponding FFT spectrum. **b** ABF-STEM image of doped SnSe with an array of darker features induced by strain. **c** ADF-STEM image of a representative strained region, and **d** enlarged ADF-STEM image showing a distorted region in the lattice. **e** Intensity profile from boxed region in panel (**d**) (from left to right). **f1**, **f2** Boxed region indicated in panels (**a**) and (**c**), respectively, showing enhancement in symmetry and decrease in θ after Te/Mo co-doping, overlaid by Sn and Se atom models. Double arrows indicate the distance between the farthest-neighboring Se-Sn pair. **g1**, **g2** IFFT image using only (002) reflections and (400) reflections from panel (**d**), respectively. **h1**, **h2** GPA of panel (**d**).

from only (020) planes (Fig. 5g1) and (400) planes (Fig. 5g2), and only (020) planes shows perturbation around the distorted region. The decrease in θ is achieved by the contraction between the farthest-neighboring Se-Sn pairs perpendicular to (020) planes (indicated by double arrows in Fig. 5f1, 2), and such contraction is compensated by the insertion of extra planes parallel to (020) planes around the highly distorted region (Fig. 5g). Geometric phase analysis (GPA) further shows the compressive nature at the extra plane insertion position (Fig. 5h1, h2). Moreover, the presence of strain fields revealed by GPA also indicates increased phonons scattering events caused by Mo substitution and lattice distortion.

### Thermal transport properties

Te and Mo are able to effectively suppress the thermal conductivities in $n$-type SnSe crystals as well. Fig. 6a depicts the total thermal conductivity ($\kappa_{tot}$) as a function of temperatures. The downtrend of $\kappa_{tot}$ is observed in the temperature range of 300−723 K with Te and Mo alloying. The $\kappa_{tot}$ decreases from ~1.8 W m⁻¹ K⁻¹ in SnSe to ~1.6 W m⁻¹ K⁻¹ in SnSe-0.75%Te, further dopes to ~1.4 W m⁻¹ K⁻¹ in SnSe-0.75%Te-0.7% Mo at 300 K. The minimum $\kappa_{tot}$ is suppressed to ~0.6 W m⁻¹ K⁻¹ in SnSe-0.75%Te-0.7%Mo at 723 K. The lattice thermal conductivity ($\kappa_{lat}$) shows a similar trend with $\kappa_{tot}$ as depicted in Fig. 6b. The room temperature $\kappa_{lat}$ is ~1.1 W m⁻¹ K⁻¹ in SnSe-0.75%Te-0.7%Mo, which is much lower than that of ~1.6 W m⁻¹ K⁻¹ in SnSe. The detail of the thermal transport is shown in Supplementary Figs. S3d–h and S4d–h, including the $\kappa_{tot}$, $\kappa_{lat}$, and other thermal transport parameters of all samples, and the heat capacity is shown in Supplementary Fig S5.

To reveal the cause of the low $\kappa_{lat}$, the Callaway model is utilized to show the influence of the point defects on the thermal conductivity[53], and the detailed calculation process is shown in Supplementary Information. As shown in Fig. 6c, the experimental data closely align with the calculated theoretical line, indicating that the reduced thermal conductivity originates from the point defects[54]. The significant strain and mass fluctuation induced by Te$_{Se}$ and Mo$_{Sn}$ strongly scatters

the phonon transport in SnSe. Meanwhile, the GPA in Fig. 5h1 and 5h2 also prove the strong strain in the distorted lattice after Te and Mo addition, contributing to the low thermal conductivity. To further comprehend the blocked phonon transport, we performed the DFT calculations to obtain the phonon dispersions as shown in Fig. 6d. The calculations are based on the refined results of SnSe and SnSe-0.75% Te-0.7%Mo at 300 K from SR-XRD, which accurately reflect the actual situation of phonon. Here, we focus on the region of Γ-X and Γ-Y, corresponding to the in-plane properties of SnSe. The slopes of the acoustic branches become lower after Te and Mo alloying, which is in harmony with the reduced sound velocity in Supplementary Table S1. The cutoff frequency is also limited to a lower value. Meanwhile, the frequencies of optical branches are also lowered, denoting the soft optical branches and the strong coupled acoustic-optical mode[55,56]. The above factors collectively contribute to the low thermal conductivity in SnSe-Te-Mo.

### Performance evaluation of SnSe-Te-Mo

The quality factor ($B$), serving as an indicator of the thermoelectric properties and determined by $B = \frac{9\mu_W}{\kappa_{lat}}\left(\frac{T}{300}\right)^{2.5}$, is depicted in Fig. 7a[57]. The addition of Te increases the $B$ factor over the whole temperature range, and the addition of Mo increases it further. Compared with pure SnSe, the $B$ factor of SnSe-0.75%Te-0.7%Mo is increased by ~217% at 300 K and ~113% at 723 K. As discussed in Fig. 1c, the final $ZT$ increases to ~0.6 at 300 K and reaches ~ 1.0 at 573 K in SnSe-0.75%Te-0.7%Mo. Fig. 7b shows the average $ZT$ ($ZT_{ave}$), which is boosted from ~0.39 in SnSe to ~0.62 in SnSe-0.75%Te, and finally rises to ~0.89 in SnSe-0.75% Te-0.7%Mo. Furthermore, we conducted two cycles of heating and cooling tests on the sample as shown in Supplementary Fig. S16, which demonstrated good thermal stability.

The single-leg thermoelectric devices based on the $n$-type SnSe crystals in this work were fabricated and their conversion efficiencies ($\eta$) were measured by the Mini-PEM, the detail of which was shown in Methods. The maximum $\eta$ is ~2.5% in SnSe under the $\Delta T$ of 300 K

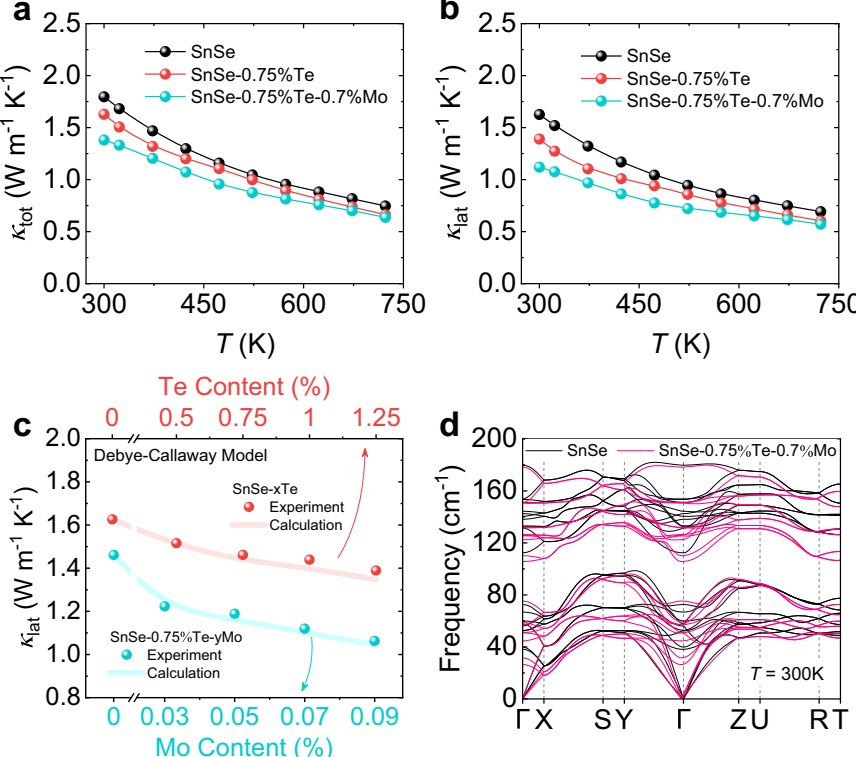

**Fig. 6 | The thermal transport properties of SnSe, SnSe-0.75%Te, and SnSe-0.75%Te-0.7%Mo. a** Total thermal conductivity as a function of temperatures. **b** Lattice thermal conductivity as a function of temperatures. **c** Callaway model simulation and the lattice thermal conductivity at 300 K of SnSe-xTe and SnSe-0.75%Te-yMo samples. **d** Phonon dispersions of SnSe and SnSe-0.75%Te-0.7%Mo.

(Supplementary Fig. S9d), while it increases to ~3.6% in SnSe-0.75%Te (Supplementary Fig. S9g), and finally it rises to ~5.3% in SnSe-0.75%Te-0.7%Mo (Fig. 7c). The relative parameters, including the output voltage, output power density, and efficiency, are provided in Supplementary Fig. S9. Moreover, as shown in Fig. 7d, the SnSe-0.75%Te-0.7% Mo possesses a superior efficiency compared to other *n*-type materials[58–65], signifying the great power generation potential of *n*-type SnSe crystals.

### Outlook of *n*-type SnSe crystals

Thermoelectric devices require comparable performance between *n*-type and *p*-type materials. Therefore, to advance the development of the *n*-type SnSe crystal and match it with the high-performance *p*-type one, we compare the thermoelectric transports in *n*-type and *p*-type SnSe for guidance. First, the doping of Na in SnSe is more effective than that of halogen. The carrier concentration can reach ~$4 \times 10^{19}$ cm$^{-3}$ in Na-doped SnSe[66], which is approximately 4 to 5 times that of Br doping and much higher than Cl and I doping[28,48,67]. Therefore, cations with high valence states are expected to be introduced to *n*-type SnSe. Some efforts have already been made, such as La[68], Ce[69], and W[70], but more effective elements are needed. Second, there are 6 valence bands in SnSe and 3 or 4 of them can participate in *p*-type electrical transport[66], while only 2 conduction bands can be utilized for *n*-type electrical transport[67]. Meanwhile, the synglisis including the momentum and energy alignment of valence bands synergistically optimize the $m^*$ and $\mu$ in *p*-type SnSe[31]. In contrast, strategies of band sharping and resonant level inducing can be conducted in *n*-type SnSe to achieve a similar effect as synglisis.

When considering the practical application of *n*-type SnSe, production costs and device manufacturing technology should be taken into account. The current process (Bridgman method) is time-consuming and can not gaurantee the quality of the crystals. New synthesis techniques for the rapid preparation of high-quality crystals can be an attractive research area. Besides, interstitial atoms can be introduced to enhance the interlayer binding force and thereby mechanical strength. The suitable contact materials also deserve attention in order to maximize the performance of *n*-type SnSe crystals in thermoelectric devices.

## Discussion

In this work, we focused on the in-plane thermoelectric performance of *n*-type SnSe crystals. Superior thermoelectric performance is achieved in SnSe-0.75%Te-0.7%Mo through symmetry modification. Firstly, the two extrinsic elements Te and Mo were successively introduced into *n*-type SnSe crystals and continuously enhanced the structural symmetry. This enhancement leads to a significant increase in carrier mobility, reaching ~422 cm$^2$ V$^{-1}$ s$^{-1}$ at 300 K benefiting from the high symmetric crystal structure. Secondly, Te alloying not only improves symmetry but also promotes band convergence. Both conduction bands participate in the electrical transport and further elevate the power factor to ~28 μW cm$^{-1}$ K$^{-2}$ at 300 K. Thirdly, the introduction of Te and Mo creates point defects in SnSe. The acoustic and optical branches are softened simultaneously therefore the phonon transport is strongly scattered. The lattice thermal conductivity is noticeably reduced, which is ~1.1 W m$^{-1}$ K$^{-1}$ at 300 K and ~0.6 W m$^{-1}$ K$^{-1}$ at 723 K. Combined with the well-optimized power factor and strongly blocked phonon transport, a high-ranged thermoelectric property is obtained in *n*-type SnSe crystals, with the room-temperature *ZT* of ~0.6 and average *ZT* of ~0.89 from 300 K to 723 K. Furthermore, we realized a single-leg conversion efficiency of ~5.3% under the temperature gradient of 300 K, which is superior in current *n*-type thermoelectric materials.

The successful application of symmetry modification in this work confirms the significance of crystal structural management in the physical and chemical property optimization of materials. The strategy

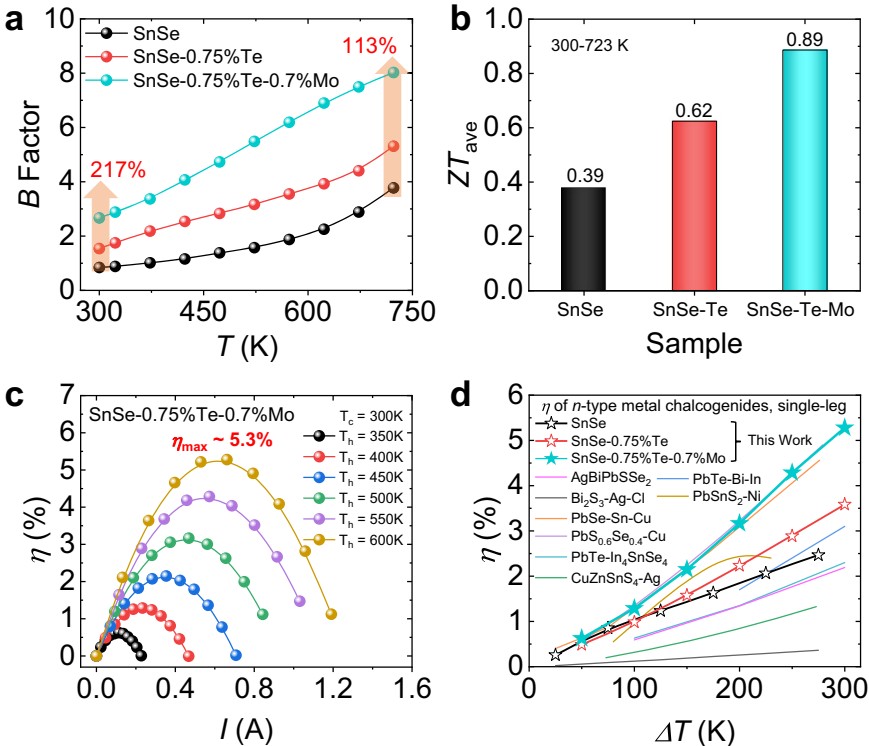

**Fig. 7 | Performance evaluation of SnSe, SnSe-0.75%Te, and SnSe-0.75%Te-0.7% Mo. a** Quality factor ($B$) as a function of temperatures. **b** The average $ZT$ ($ZT_{ave}$) at 300–723 K. **c** Single-leg conversion efficiencies ($\eta$) of SnSe-0.75%Te-0.7%Mo as functions of current at different hot-end temperatures ($T_h$), with the constant cold-end temperature ($T_c$) of ~ 300 K. **d** The comparison of $\eta$ between samples in this work and other $n$-type thermoelectric materials[58–65].

of adjusting the crystal structure, especially symmetry, can be extended to other thermoelectric material systems. Depending on the characteristics of the material, one can either enhance symmetry or break local symmetry to balance the electrical and heat transport. Besides, the optimized $n$-type SnSe crystal in this work is promised to be applied in both renewable electricity generation and thermoelectric cooling, particularly when matched with the high-performance $p$-type counterpart. The next crucial step is to identify a suitable contact material for SnSe to maximize its performance. The design of the contact interface and device geometry will further advance the commercial application of SnSe.

## Methods
### Sample synthesis
Raw materials, including Sn (bulk, 5 N), Se (shot, 4 N), Te (bulk, 5 N), Mo (powder, 3 N), and $SnBr_2$ (powder, 2 N), were used to synthesize the series of $SnSe_{0.97-x}Te_xBr_{0.03}$ (x = 0, 0.5%, 0.75%, 1%, 1.25%) and $Sn_{1-y}Mo_ySe_{0.9625}Te_{0.0075}Br_{0.03}$ (y = 0, 0.3%, 0.5%, 0.7%, 0.9%) crystal samples. To note, all samples in this work were doped with 3% Br to ensure the $n$-type transport, and for simplicity, Br was not indicated in the nominal ingredients below. Stoichiometric amounts of high-purity raw materials were weighed and mixed into the vacuum-sealed quartz tubes. The tubes were heated to 1313 K for 16 h and kept at 1313 K for 16 h to obtain the SnSe ingots. Then the ingots were ground into powders and sealed into tailor-made quartz tubes, whose bottoms were pointed for the growth of the seed crystal. The tubes were loaded into the verticle furnace. The middle part of the glass tube was in the heated area and the temperature of the bottom was ~50 K lower than the heated area, leading to a temperature gradient. The furnace was heated to ~1313 K for 16 h and then the temperature slowly dropped to ~1073 K at the rate of ~1 K h⁻¹. Finally, the furnace cooled to room temperature, and the SnSe crystals were obtained.

### Electrical transport properties
The n-type SnSe crystals were cut and polished into bars with the dimensions of ~3 × 3 × 8 mm³ for the measurement of Seebeck coefficient and electrical conductivity by the ZEM-3 instrument (Advance Riko, Japan). The measurements were carried out in a helium atmosphere at 300–723 K. The samples were coated with a thin layer of boron nitride to prevent possible evaporation. The uncertainty of the Seebeck coefficient and electrical conductivity measurement is within 5 %.

### Thermal transport properties
The thermal conductivity was calculated by $\kappa_{tot} = D\rho C_p$. $D$ is the thermal diffusivity, which was measured by the LFA457 instrument (Netzsch, Germany) in the nitrogen atmosphere from 300 K to 723 K, and analyzed by the Cowan model with pulse correction. The samples for testing were the pieces cut and polished into the size of ~6 × 6 × 1.5 mm³. The uncertainty of $D$ is within 5%. The density ($\rho$) was determined by the mass and dimensions of the sample. $C_p$ represents the specific heat capacity estimated by the Debye model.

### Hall measurement
The Hall coefficient was measured by the Lake Shore 8400 Hall measurement system (Lake Shore Cryotronics, USA) under the magnetic field of 1 T. The crystal samples were cut and polished into pieces of ~8 × 8 × 0.8 mm³ for measurement. The carrier concentration ($n$) was calculated by $n = 1/(e \cdot R_H)$, where e is the electron charge. The carrier mobility ($\mu$) is calculated by $\mu = \sigma \cdot R_H$, where $\sigma$ is the electrical conductivity.

### Bandgap measurement
The optical bandgap was obtained by the optical diffuse reflectance measurements at room temperature using the UV-3600 Plus UV–VIS NIR spectrophotometer (Shimadzu, Japan). The samples were ground

into powders for measurement with $BaSO_4$ as the standard (100% reflectance). The generated reflectance versus wavelength data was used to estimate the bandgap by converting reflectance to absorption data according to the Kubelka-Munk equation: $\alpha/S = (1-R)^2/2R$, where $\alpha$, $S$ and $R$ are the reflectance, and $\alpha$ and S are the absorption coefficients, scattering coefficients and reflectance, respectively.

## X-ray diffraction

The small piece peeled from the crystal sample along the cleavage plane was used for phase constitution identification. The diffraction patterns were characterized by the FRINGE CLASS X-ray diffraction (LANScientific, China) with Cu $K_\alpha$ radiation ($\lambda \sim 0.15418$ nm). The measurement was conducted at 40 kV and 20 mA with a scanning speed of 6 degrees per minute.

## Synchrotron radiation X-ray diffraction

The SR-XRD data are obtained at BL14B1 of the Shanghai Synchrotron Radiation Facility (SSRF) using X-ray with a wavelength of 0.6887 Å. The samples were ground into powders and sifted out through 400 mesh screens, then seeled into the quartz capillary tubes with the $N_2$ atmosphere. The tubes were heated from 300 K to 723 K at the rate of 1.5 K $min^{-1}$. GSAS software was used to analyze the test data and the accurate structural information of SnSe samples in this work was obtained.

## X-ray photoelectron spectroscopy

The chemical valences and compositions of SnSe-0.75%Te-0.7%Mo were analyzed using an ESCALAB 250Xi X-ray photoelectron spectrometer (Thermo Fisher Scientific Inc., America) with an Al $K_\alpha$ radiation source (1486.6 eV photon energy, 300 W) at $10^{-19}$ Pa pressure.

## Conversion efficiency measurement

The single-leg conversion efficiency of the samples was carried out by the thermoelectric conversion efficiency measurement system Mini-PEM (Advance Riko, Japan). Ni was electroplated on the surface of the samples and the copper sheet was used as electrodes, connecting by the commercial silver paste. The silicone grease was applied between the copper and the measurement instrument to ensure the heat conducting. The cold-side temperature $T_c$ was kept at ~300 K and the hot-side temperature $T_h$ was set from 350 K to 600 K.

## Structure characterization

The microstructural analyses were performed using a spherical aberration-corrected transmission electron microscope (JEM-ARM200F NEOARM, JEOL Ltd.) equipped with a cold field emission gun (FEG) operating at an accelerating voltage of 200 kV. The instrument was additionally configured with an energy-dispersive X-ray spectroscopy (EDS) system for elemental mapping. Specimen preparation was carried out following standard metallographic procedures, including precision cutting, mechanical grinding, and polishing, followed by final thinning using Ar-ion milling (PIPS, Gatan Inc.) with liquid nitrogen cooling to minimize thermal damage to the sample.

## Data availability

The authors declare that the data supporting the findings of this study are available on reasonable request.

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

## Acknowledgements

This work was supported by the National Natural Science Foundation of China (52450001 and 523B2007), the Tencent Xplorer Prize, the National Natural Science Foundation of China (22409014, 12374023, and 524B2004), the Beijing Natural Science Foundation (JQ18004), the

111 Project (B17002), National Key Research and Development Program of China (2023YFB3809400), the China National Postdoctoral Program for Innovative Talents (BX20230456), China Postdoctoral Science Foundation (2024M754057) and the Young Scientists Fund of the National Natural Science Foundation of China (No. 52403348). T.Z. is supported by the National Natural Science Foundation of China (22308376), the Outstanding Young Scholars Foundation of China University of Petroleum (Beijing) (2462023BJRC015). We thank BL14B1 and the user experiment assist system (Shanghai Synchrotron Radiation Facility, SSRF) for the SR-XRD experiments, the high performance computing (HPC) resources at Beihang University.

## Author contributions

L.-D.Z. and Q.T. conceived the idea, designed the experiments, and supervised the research. H.S. performed the sample synthesis and property measurements and wrote the manuscript. Y.W., Z.H., and X.G. conducted the microstructure observation. S.B. carried out the theoretical calculations. L.S. performed the XRD measurement and assisted in completing the data analysis. C.C., B.Q., Y.Q., and T.Z. assisted in completing the data analysis. T.G. participated in the property measurement. D.L. fabricated the single-leg device. S.L. performed the SR-XRD measurement. H.L. and X.Q. carried out the XPS measurement. All authors analyzed the results and commented on the manuscript.

## Competing interests

The authors declare no competing interests.
