## [Transparent Peer Review file · Nature Communications]

Crystal symmetry modification enables high-ranged in-plane thermoelectric performance in n-type SnSe crystals

Corresponding Author: Professor Li-Dong Zhao

Version 0:

Reviewer comments:

Reviewer #1

(Remarks to the Author)

Authors investigated doped SnSe materials for thermoelectric. Analyzed electrical and thermal transport supported by various characterizations including DFT. They claimed change in crystal symmetry played a major role to achieve high zT in SnSe in-plane. There are some important suggestions and points need to be revise, given as below.

1. Authors previous work has already claimed to have zT of 1 for n-type SnSe in-plane. How current work adding benefit to TE. Because at the end, similar response after both processes observed by authors and other researcher too. Also, if out-plane showed outstanding performance of 2.5 or more, what is the benefit to have in-plane, other than mechanical properties.

2. It would be great if authors provide lattice strain changes after Te-Mo doping.

3. My concern about electrical conductivity, decreasing at broader temp range to 50-70 S/cm, which could be both due to electron-phonon scattering and low carrier concentration order. Over the time such response gets degraded and zT went down more. Thus, it is always advisable to have high carrier concentration for TE. Also, both studied compositions in Figure S2 and S3 have very marginal difference in terms of zT.

4. Authors provide a logic, "resulting in fewer free electrons in the matrix and accounting for the decreased carrier concentration". It cannot be ruled out that minority carrier increases after Mo alloying, thereby, decreasing carrier concentration was observed.

5. As Mo alloying significantly boost mobility in SnSe in-plane, how about out-plane mobility? does it also have same values?

6. In principle, high symmetry or nearby atomic position improves a strong network for phonon transport. After Mo addition, if phonon scattering is more, then why mobility not harmed with it, electron-phonon scattering picture become bit complex.

7. Why author do not try to make single/multi couple device, as they are aware of p-type SnSe, let's see how it performs.

8. Author used Mo, and denoted Pb in sample synthesis section. Sn_{1-x}Pb_xSe_{0.97}Br_{0.03}

9. Do authors identified Br by any methods, what is the role that Br is contributing? In first series no dopant, then mix of Te and Br and then Mo. So, authors neglected Br effect.

10. Authors should consider, Br also acts well, because Cl, Br, I these are halogens elements that have already put its significance towards band convergence and modification when there are nearby dopant atoms other than matrix elements.

11. Ideally grown crystal shows high zT values, but when it transforms to consider SPSe/Hot pressed its performance get suppressed down. For device perspective, TE community need viable synthesis techniques.

12. I do not hear about GASA software, do author means GSAS for Rietveld refinement?

13. Change in ΔE one can see for any dopant in electronic structure calculation, so particular element is doing great cannot be a true statement.

14. Does author feel single parabolic model valid here, as they are focusing more on band convergence and crystal symmetry change, why SPB fits here? Strong dependence of the mobility on the band masses may resulting differs between a SPB and 2PB system. Also, possibility of 2PB cannot be denied. <https://doi.org/10.1016/j.jmat.2020.10.013>

15. Figure 6d, comparison should be rechecked, I see some literature have 5-8% efficiency, which is above to this work. one example, <https://www.nature.com/articles/s41467-024-48268-3>

Reviewer #2

(Remarks to the Author)

The authors report on the effect of Te for Se and Mo for Sn substitution on the thermoelectric properties of samples grown by

Bridgman method. The results are somehow interesting; however the present data does not allow to establish/confirm if the composition changes really affect the structure and crystal symmetry, as claimed by the authors. My comments are :

- 1) It is quite surprising to see that the Bridgman samples are only characterized by powder XRD technique (cleavage plane or powder), especially considering that the work is focused on the crystal structure. It is most probably very easy to extract single crystals from the Bridgman samples and to perform single crystal XRD analysis. This can provide much more precise and detailed information on the crystal structure, including cell parameters, bonds, angles, Basis (& aniso), occupancies, ...
- 2) From all the PXRD refinements (synchrotron or not), and for each composition and T, detailed structural parameters must be provided: Basis, occupancies, reliability factors, angles distances etc....
- 3) structural data from all the compositions are not provided. The changes of cell parameters are only given for three compositions. The authors must concentrate their work on RT data and to determine if there is a linear change of cell parameters, bond angles versus x and y. It is very surprising that this has not been performed and presented in the present article. Long acquisition monochromated powder XRD data must be recorded for each composition.
- 4) EDS mapping must be performed on each composition to attest the purity of the samples.

After this in-depth structural analysis is performed, then the article can be resubmitted.

as a general comment, I don't think that the term of high symmetry is appropriate. Even the cell parameters are changed, the symmetry itself is not changed, as the space group is unchanged. there is no specific disorder induced by the anionic or cationic substitution which modify the symmetry. Then we can wonder if the change of properties is really affected by the change of symmetry. The lattice constrains and the final stoichiometry should be also considered.

One more comment: it is not specified in which direction of the samples, the electrical and thermal properties are measured. Is the texture similar for all the samples? this must be presented and discussed as the texture can strongly influence the transport properties. from only the XRD data on cleavage surfaces, it is impossible to determine the strength of the texture.

Reviewer #3

(Remarks to the Author)

This study presents the critical role of Te and Mo alloying induced crystal symmetry modifications in SnSe. Improved symmetry results in converging conduction bands with increased carrier mobility ($\sim 422 \text{ cm}^2 \text{ V}^{-1} \text{ s}^{-1}$). Lattice thermal conductivity is reduced to $\sim 1.1 \text{ W m}^{-1} \text{ K}^{-1}$ due to soft acoustic and optical phonon branches, with $zT \sim 0.6$ at 300 K, with an average ZT of ~ 0.89 over the range 300–723 K. A single-leg thermoelectric device based on this material achieves $\sim 5.3\%$ efficiency for a temperature gradient (ΔT) of $\sim 300 \text{ K}$. Few inconsistencies as enumerated below requires a moderate revision prior to publication.

1. Clarify the dopability and solubility limits of Te and Mo in SnSe.
2. How do the lattice parameters of SnSe vary with increasing Mo content and decreasing Te content? In addition to discussing the optimal sample, the results for other Mo- and Te-based samples can also be included.
3. The synthesized crystal appears to be polycrystalline and is not well characterized or described in terms of its crystallinity. The single-crystal SnSe demonstrates impressive ZT values ranging from 2.2 to 2.6 at 913 K. In contrast, polycrystalline versions of SnSe, synthesized through various methods, exhibit significantly lower overall ZT values, which can be compared to elucidate the relevance of crystal symmetry and grain boundary effects.
4. The continuous phase transition of SnSe from a low-symmetric Pnma to a high-symmetric Cmcm phase between 600 K and 800 K may affect the long-term stability and reliability of SnSe-based thermoelectric devices. Justify the thermal stability of the synthesized samples.
5. While the study reports improvements in n-type SnSe, the achieved power factor ($\sim 28 \mu\text{W cm}^{-1} \text{ K}^{-2}$) and ZT (~ 0.6) at 300 K are still relatively low compared to state-of-the-art p-type SnSe materials, which have shown power factors up to $\sim 85 \mu\text{W cm}^{-1} \text{ K}^{-2}$ and ZT of ~ 1.4 at ambient temperature. This disparity in performance between n-type and p-type SnSe needs to be emphasized more clearly in results and discussion.
6. The temperature-dependent behavior of the heat capacity and its role in influencing the thermal transport properties across the studied temperature range, needs to be established well.
7. What is the grain size? The lack of structural characterization raises concerns about elemental distribution and sample stoichiometry.
8. There are prevailing typos and errors which needs to be thoroughly checked and corrected.

Version 1:

Reviewer comments:

Reviewer #1

(Remarks to the Author)

Authors discussed answers nicely, I recommend publishing in Nature Comm.

Reviewer #2

(Remarks to the Author)
the authors clearly addressed all of my comments.

Reviewer #3

(Remarks to the Author)
The authors have addressed the questions and considerations, enhancing the clarity of their research. Therefore, the manuscript is recommended for acceptance.

Referee 1:

General Comment: Authors investigated doped SnSe materials for thermoelectric. Analyzed electrical and thermal transport supported by various characterizations including DFT. They claimed change in crystal symmetry played a major role to achieve high zT in SnSe in-plane. There are some important suggestions and points need to be revise, given as below.

Response: Thanks for the critical comments and useful suggestions. We have provided a point-to-point response as follows and we hope it can resolve your doubts.

Detailed Comment 1: Authors previous work has already claimed to have zT of 1 for n -type SnSe in-plane. How current work adding benefit to TE. Because at the end, similar response after both processes observed by authors and other researcher too. Also, if out-plane showed outstanding performance of 2.5 or more, what is the benefit to have in-plane, other than mechanical properties.

Response: This question is highly valuable for the development of n -type SnSe crystals. In our previous work, the maximum in-plane ZT has reached 1 in n -type SnSe crystals by Pb alloying, however, its ZT is not satisfying around near room temperature. In the current work, the ZT values have a significant increase in the mid-to-low temperature range, especially at room temperature, reaching up to 0.6. Meanwhile, the samples in this work are Pb-free, which avoids the potential environmental and toxic issues during application.

The n -type SnSe crystal has greater application potential along the in-plane direction. Except for the mechanical properties, the in-plane direction is more convenient for electroplating and welding, which can facilitate device manufacturing. Additionally, the in-plane samples exhibits inherently higher carrier mobility than the out-of-plane

samples, making it more applicable around near room temperature. (doi: 10.1126/science.ade9645) We have added more discussions about the difference between in-plane and out-of-plane in our manuscript.

Revision: Our emphasis was evaluating the in-plane thermoelectric performance, as illustrated in Fig. S1b. Although *n*-type SnSe crystals have performed excellent thermoelectric performance along the out-of-plane direction, the cleaved characteristic makes it difficult for application. In contrast, the in-plane direction has advantages in mechanical strength, weldability, and electroplatability, which is more suitable for device manufacturing.

Fig. S1 The photo and schematic diagram of the SnSe crystal. a The photo of the SnSe crystal cleaved along (100) plane. **b** The schematic diagram of the SnSe crystal which shows how to cut out samples for thermoelectric transport measurement.

Detailed Comment 2: It would be great if authors provide lattice strain changes after Te-Mo doping.

Response: Thanks for your good suggestion. The lattice strain can reflect the state of the crystal lattice after alloying and is associated with low thermal conductivity. We observed the microstructure and utilized geometric phase analysis (GPA) to show the lattice strain. The relevant content has been added to the manuscript.

Revision: Moreover, the presence of strain fields revealed by GPA also indicates increased phonons scattering events caused by Mo substitution and lattice distortion.

Meanwhile, the GPA in Fig. 5h1 and 5h2 also prove the strong strain in the distorted lattice after Te and Mo addition, contributing to the low thermal conductivity.

Fig. 5 AC-STEM characterization of SnSe before and after Te/Mo co-doping. **a** ADF-STEM image of pristine SnSe viewed along [010] direction, inset showing the corresponding FFT spectrum. **b** ABF-STEM image of doped SnSe with an array of darker features induced by strain. **c** ADF-STEM image of a representative strained region, and **d** enlarged ADF-STEM image showing a distorted region in the lattice. **e** Intensity profile from boxed region in panel **d** (from left to right). **f1, f2** Boxed region indicated in panels **a** and **c**, respectively, showing enhancement in symmetry and decrease in θ after Te/Mo co-doping, overlaid by Sn and Se atom models. Double arrows indicate the distance between the farthest-neighboring Se-Sn pair. **g1, g2** IFFT image using only (002) reflections and (400) reflections from panel **d**, respectively. **h1, h2** GPA of panel **d**.

Detailed Comment 3: My concern about electrical conductivity, decreasing at broader temp range to 50-70 S/cm, which could be both due to electron-phonon scattering and low carrier concentration order. Over the time such response gets degraded and zT went down more. Thus, it is always advisable to have high carrier concentration for TE. Also, both studied compositions in Figure S2 and S3 have very marginal difference in terms of zT .

Response: Your concern is valuable and worth discussing. In our work, we pursue a saturated ZT curve to realize a wide working temperature range, instead of the peak value of ZT . Therefore, the high carrier concentration is not always beneficial for thermoelectric properties. We intentionally control the carrier concentration appropriately to achieve higher carrier mobility. (doi: 10.54227/mlab.20220004) Moreover, excellent thermoelectric materials are always heavy-doped semiconductors, which perform the metal-like conducting characteristic. However, the thermal conductivity also decreases with the temperature rising, compensating for the degradation induced by reduced electrical conductivity, and the final ZT does not drop significantly.

Fig. S3 and S4 (corresponding to Fig. S2 and Fig. S3 before modification.) show the thermoelectric properties of SnSe-Te and SnSe-Te-Mo respectively. The solid black

lines in Figure S4 represent the best-optimized sample in Figure S3, and a significant difference can be observed between them.

Detailed Comment 4: Authors provide a logic, "resulting in fewer free electrons in the matrix and accounting for the decreased carrier concentration". It cannot be ruled out that minority carrier increases after Mo alloying, thereby, decreasing carrier concentration was observed.

Response: Thank you for pointing out another possible variation in carrier concentration. In our point of view, it is inappropriate to attribute the decrease in carrier concentration to an increase in minority carriers in this case. As is known, the Seebeck coefficient can be defined as $S_{\text{tot}} = (\sigma_n S_n + \sigma_p S_p) / (\sigma_n + \sigma_p)$, where σ_n , σ_p , S_n , and S_p are σ for n - and p -type materials, and S for n -type and p -type materials respectively. Once the minority carriers increase, the total S will be degraded. However, the S after Mo alloying in this work does not show the downward trend in Fig. 2b. Moreover, Mo is determined to be in the +2 oxidation state according to XPS. The isoelectronic substitution of Sn²⁺ appears not to cause an increase in minority carriers (holes) in this work. Therefore, we conclude that the decreased carrier concentration should not be caused by the change in minority carriers.

Detailed Comment 5: As Mo alloying significantly boost mobility in SnSe in-plane, how about out-plane mobility? does it also have same values?

Response: Thanks for your good suggestion. We have supplemented the data of the electrical conductivity and carrier mobility along the out-of-plane in Tab S5. Their values are not the same due to the different transport characteristics between in-plane and out-of-plane. However, they perform a similar trend, showing minor changes after Te alloying and a significant increase after Mo alloying.

Revision: (In the manuscript) In addition, we also considered the variation in out-of-plane mobility and found that it exhibits a similar trend as that along the in-plane direction as shown in Tab. S5, demonstrating that symmetry enhancing can promote electrical transport properties along both directions, rather than being specific to any particular direction.

(In the Supplementary Information)

Table S5 The comparison of the electrical conductivity and carrier mobility along the two directions at room temperature. The σ_{in} , σ_{out} , μ_{in} , and μ_{out} are the electrical conductivity and carrier mobility along the in-plane and out-of-plane, respectively.

Sample	σ_{in}	μ_{in}	σ_{out}	μ_{out}
SnSe	346.39	247.42	139.55	99.68
SnSe-0.75%Te	487.60	277.05	171.79	97.61

Detailed Comment 6: *In principle, high symmetry or nearby atomic position improves a strong network for phonon transport. After Mo addition, if phonon scattering is more, then why mobility not harmed with it, electron-phonon scattering picture become bit complex.*

Response: Thanks for your good suggestion. The factors affecting electron transport are diverse, and we need to focus on the principal contradiction. The crystal symmetry can strongly affect carrier mobility by changing the relaxation time. (doi: 10.1039/d0tc03270k) The primary role of Mo in this work is to enhance the crystal symmetry, and thus it can be considered a key factor in improving carrier mobility. Therefore, the carrier mobility can overcome the adverse effects of electron-phonon scattering and shows an increasing trend.

Detailed Comment 7: *Why author do not try to make single/multi couple device, as they are aware of p-type SnSe, let's see how it performs.*

Response: Thanks for your good suggestions. We tried to make a 7-pair thermoelectric device based on *n*-type and *p*-type SnSe crystals, and its performance is not satisfactory, as shown in the following figure. Although SnSe crystals perform well in thermoelectric properties, the SnSe-based device fabricating still confronts some challenges. First, the thermoelectric performance of *n*-type SnSe can not match with the *p*-type one. The properties of materials can not be fully activated within the device because of the mismatch. Second, we think that the single contact material cannot be simultaneously applied to both *n*-type and *p*-type SnSe owing to its wide bandgap ($\sim 0.86\text{eV}$). Therefore, a new and suitable contact material is needed especially for *n*-type SnSe. Meanwhile, the continuous phase transition of SnSe and the resulting expansion or contraction behavior impose stringent requirements on the thermal expansion of the contact materials. It exacerbates the difficulty in contact material development. Certainly, we are also committed to the application of SnSe crystals and have provided a perspective on the development of SnSe crystals and devices.

Revision: Thermoelectric devices require comparable performance between n -type and p -type materials. Therefore, to advance the development of the n -type SnSe crystal and match it with the high-performance p -type one, we compare the thermoelectric transports in n -type and p -type SnSe for guidance. First, the doping of Na in SnSe is more effective than that of halogen. The carrier concentration can reach $\sim 4 \times 10^{19} \text{ cm}^{-3}$ in Na-doped SnSe, which is approximately 4 to 5 times that of Br doping and much higher than Cl and I doping. Therefore, cations with high valence states are expected to be introduced to n -type SnSe. Some efforts have already been made, such as La, Ce, and W, but more effective elements are needed. Second, there are 6 valence bands in SnSe and 3 or 4 of them can participate in p -type electrical transport, while only 2 conduction bands can be utilized for n -type electrical transport. Meanwhile, the synglisis including the momentum and energy alignment of valence bands synergistically optimize the m^* and μ in p -type SnSe. In contrast, strategies of band sharpening and resonant level inducing can be conducted in n -type SnSe to achieve a similar effect as synglisis.

When considering the practical application of n -type SnSe, production costs and device manufacturing technology should be taken into account. The current process (Bridgman method) is time-consuming and can not guarantee the quality of the crystals. New synthesis techniques for the rapid preparation of high-quality crystals can be an attractive research area. Besides, interstitial atoms can be introduced to enhance the interlayer binding force and thereby mechanical strength. The suitable contact materials also deserve attention in order to maximize the performance of n -type SnSe crystals in thermoelectric devices.

Detailed Comment 8: Author used Mo, and denoted Pb in sample synthesis section.

Sn_{1-x}Pb_xSe_{0.97}Br_{0.03}

Response: Thanks for your good suggestion. We sincerely apologize for this clerical error and we have corrected it.

Revision: Raw materials, including Sn (bulk, 5N), Se (shot, 4N), Te (bulk, 5N), Mo (powder, 3N), and SnBr₂ (powder, 2N), were used to synthesize the series of SnSe_{0.97-x}Te_xBr_{0.03} ($x = 0, 0.5\%, 0.75\%, 1\%, 1.25\%$) and Sn_{1-y}Mo_ySe_{0.9625}Te_{0.0075}Br_{0.03} ($y = 0, 0.3\%, 0.5\%, 0.7\%, 0.9\%$) crystal samples.

Detailed Comment 9: Do authors identified Br by any methods, what is the role that Br is contributing? In first series no dopant, then mix of Te and Br and then Mo. So, authors neglected Br effect.

Response: The role of Br is a fundamental issue for n-type SnSe. Br acts as a donor dopant and can convert SnSe from the intrinsic *p*-type to the *n*-type. The *n*-type transport characteristic (the black line in Fig. 2b) can prove that Br has been effectively doped in SnSe crystals. Besides, Br is a widely accepted dopant of *n*-type SnSe and many works have proven that Br can effectively doped and uniformly distribute in the SnSe matrix (doi: 10.1126/science.aaq1479). For convenience, we used abbreviations in this work while discussing and we also explained it in the manuscript. In “Electrical Transport Properties” section, we said “ Herein, the Bridgman method was employed to synthesize two series of *n*-type SnSe crystal samples, namely SnSe_{0.97-x}Te_xBr_{0.03} ($x = 0, 0.5, 0.75, 1\%, 1.25\%$) and Sn_{1-y}Mo_ySe_{0.9625}Te_{0.0075}Br_{0.03} ($y = 0, 0.3, 0.5, 0.7\%, 0.9\%$), referred to as SnSe-*x*Te and SnSe-0.75%Te-*y*Mo respectively for simplicity in this work”. To avoid misunderstanding, we also added the low-magnification scanning electron microscope (SEM) images and energy dispersive spectrometer (EDS) mapping images of different elements in our work. The EDS mapping of Br confirms it is well-distributed in the matrix.

Revision: (In the manuscript) The EDS mapping was also conducted to confirm that the element distribution is uniform in samples as shown in Fig S6.

(In the Supplementary Information)

Fig. S6 Low-magnification scanning electron microscope (SEM) images and energy dispersive spectrometer (EDS) mapping images of different elements. a1 SEM image of SnSe and the corresponding elemental mappings for **a2** Sn, **a3** Se, and **a4** Br. **b1** SEM image of SnSe-0.75%Te and the corresponding elemental mappings for **b2** Sn, **b3** Se, **b4** Br, and **b5** Te. **c1** SEM image of SnSe-0.75%Te-0.7%Mo and the corresponding elemental mappings for **c2** Sn, **c3** Se, **c4** Br, **c5** Te, and **c6** Mo.

Detailed Comment 10: Authors should consider, Br also acts well, because Cl, Br, I these are halogens elements that have already put its significance towards band convergence and modification when there are nearby dopant atoms other than matrix elements.

Response: The universality and specificity of halogen elements in *n*-type SnSe is an interesting topic. However, as we explained in the question above, Br was doped into SnSe as the donor dopant to realize the *n*-type transport, as a prerequisite for this work, and the role of Br in *n*-type SnSe has been extensively discussed and comprehended. So the core issue here is to analyze the effects of Te and Mo, and it seems not necessary to discuss the role of Br.

Detailed Comment 11: Ideally grown crystal shows high zT values, but when it transforms to consider SPSe/Hot pressed its performance get suppressed down. For device perspective, TE community need viable synthesis techniques.

Response: Thanks for your good suggestion. From the device perspective, neither SnSe crystal nor polycrystal can achieve excellent thermoelectric and mechanical properties simultaneously. The viable synthesis techniques that can reduce time costs and improve crystal quality are needed for SnSe. Meanwhile, we should also notice that the crystal obtained by the Bridgman method can meet the needs of device manufacturing. (doi: 10.1126/science.adg7196) The outlook of the application on SnSe crystals has been added to the manuscript.

Revision: When considering the practical application of *n*-type SnSe, production costs and device manufacturing technology should be taken into account. The current process

(Bridgman method) is time-consuming and can not guarantee the quality of the crystals. New synthesis techniques for the rapid preparation of high-quality crystals can be an attractive research area. Besides, interstitial atoms can be introduced to enhance the interlayer binding force and thereby mechanical strength. The suitable contact materials also deserve attention in order to maximize the performance of *n*-type SnSe crystals in thermoelectric devices.

Detailed Comment 12: *I do not hear about GASA software, do author means GSAS for Rietveld refinement?*

Response: Thanks for your good suggestion. It is indeed GSAS for Rietveld refinement and we have corrected it in the manuscript.

Revision: GSAS software was used to analyze the test data and the accurate structural information of SnSe samples in this work was obtained.

Detailed Comment 13: *Change in ΔE one can see for any dopant in electronic structure calculation, so particular element is doing great cannot be a true statement.*

Response: Thanks for your good suggestion. Maybe any dopant can induce some degree of change in ΔE in electronic structure calculation, but it is unusual and noteworthy that an over 20% decrease happened after Te alloying at room temperature, from ~ 0.13 eV in Te-free SnSe to ~ 0.1 eV in Te-alloyed SnSe in Fig. 4c. We have provided a more detailed discussion about the change in ΔE in the manuscript to ensure the rigor of the article.

Revision: Fig. 4b presents a comparison between the band structures of SnSe and SnSe-0.75%Te at room temperature, and the energy difference (ΔE) of the two conduction bands decreases from ~ 0.13 eV in SnSe to ~ 0.1 eV in SnSe-0.75%Te. A reduction in ΔE by more than 20% demonstrates the degeneracy of the conduction bands.

Detailed Comment 14: *Does author feel single parabolic model valid here, as they are focusing more on band convergence and crystal symmetry change, why SPB fits here? Strong dependence of the mobility on the band masses may resulting differs between a SPB and 2PB system. Also, possibility of 2PB cannot be denied.*
<https://doi.org/10.1016/j.jmat.2020.10.013>

Response: Thanks for your good suggestion. In this work, the Pisarenko relationship based on the SPB model is employed to illustrate the increase in effective mass, while the band convergence is verified through the DFT calculations. In this case, the SPB model can be used to demonstrate the variation of the effective mass and similar approaches have also been adopted in other works. For example, in a study on SnSe polycrystals (doi: 10.1038/s41467-024-48635-0), W also converged conduction bands and thereby enlarged the effective mass, which is proven by the Pisarenko relationship

based on the SPB model and DFT calculations respectively. Similar applications of the SPB model can also be found in other material systems, like *p*-type PbSe (doi: 10.1038/s41467-022-31939-4), *p*-type GeTe (doi: 10.1002/aenm.202304029), and *n*-type GeSe (doi: 10.1016/j.nanoen.2022.107434).

Detailed Comment 15: *Figure 6d, comparison should be rechecked, I see some literature have 5-8% efficiency, which is above to this work. one example, <https://www.nature.com/articles/s41467-024-48268-3>*

Response: Thanks for your good suggestion. The comparison of efficiency should be carried out under a same/similar temperature difference (ΔT). The maximum ΔT is 300 K for the comparison in our work (with a hot-end temperature of 600 K), considering that the peak value of ZT appears at ~ 600 K in SnSe-0.75%Te-0.7%Mo (Fig. 1c). The efficiency of $\sim 5.3\%$ here has reached the highest level in *n*-type single-leg devices in the current reports. Some works report the higher efficiency because of its higher ΔT . For example, the efficiency in the work you mentioned reaches $\sim 8\%$ under the ΔT of 500 K, while it drops to $\sim 5\%$ when ΔT is 300 K.

Referee 2:

General Comment: *The authors report on the effect of Te for Se and Mo for Sn substitution on the thermoelectric properties of samples grown by Bridgman method. The results are somehow interesting; however the present data does not allow to establish/confirm if the composition changes really affect the structure and crystal symmetry, as claimed by the authors. My comments are :*

Response: Thanks for the critical comments and useful suggestions. We have provided a point-to-point response as follows and we hope it can resolve your doubts.

Detailed Comment 1: *It is quite surprising to see that the Bridgman samples are only characterized by powder XRD technique (cleavage plane or powder), especially considering that the work is focused on the crystal structure. It is most probably very easy to extract single crystals from the Bridgman samples and to perform single crystal XRD analysis. This can provide much more precise and detailed information on the crystal structure, including cell parameters, bonds, angles, Basis (& aniso), occupancies, ...*

Response: Thanks for your good suggestion. Growing SnSe crystals by the Bridgman method is a highly mature technique in thermoelectrics, and we have researched it since 2014 (doi: 10.1038/nature13184). The quality and reliability of the obtained crystals have been extensively validated. Meanwhile, precise and detailed information on the

crystal structure can be obtained by the SR-XRD, and the related data has been added to the Supplementary Information.

Revision: (In the manuscript) The detailed structural data of the three samples is also provided in Tab. S2-S4.

(In the Supplementary Information)

Table S2 The detailed refinement results of SnSe crystal with the temperature from 300 K to 723 K.

T (K)	Sn					Se				
	x	y	z	Occ	Uiso	x	y	z	Occ	Uiso
300	0.12059	0.25	0.10008	1.015	0.03	0.85606	0.25	0.49092	0.953	0.023
323	0.12054	0.25	0.09999	0.997	0.028	0.85601	0.25	0.49083	1.008	0.022
373	0.11989	0.25	0.09714	1.000	0.029	0.85536	0.25	0.48799	1.004	0.022
423	0.11922	0.25	0.09575	1.004	0.025	0.85469	0.25	0.48660	1.007	0.018
473	0.11982	0.25	0.09403	1.003	0.026	0.85529	0.25	0.48488	0.996	0.020
523	0.11879	0.25	0.08438	0.990	0.023	0.85426	0.25	0.47523	1.019	0.016
573	0.12082	0.25	0.08031	0.996	0.027	0.85629	0.25	0.47116	1.025	0.020
623	0.12026	0.25	0.07596	0.997	0.025	0.85573	0.25	0.46681	0.999	0.019
673	0.11963	0.25	0.06570	1.006	0.020	0.85509	0.25	0.45650	0.974	0.013
723	0.11968	0.25	0.06456	1.007	0.018	0.85514	0.25	0.45660	0.972	0.012

Table S3 The detailed refinement results of SnSe-0.75%Te crystal with the temperature from 300 K to 723 K.

T (K)	Sn					Se				
	x	y	z	Occ	Uiso	x	y	z	Occ	Uiso
300	0.11985	0.25	0.09425	0.994	0.026	0.85531	0.25	0.48509	1.011	0.020
323	0.11982	0.25	0.09523	0.995	0.028	0.85524	0.25	0.48607	1.010	0.021
373	0.11970	0.25	0.09642	1.009	0.030	0.85516	0.25	0.48726	1.007	0.023
423	0.11957	0.25	0.09230	0.999	0.027	0.85504	0.25	0.48314	1.016	0.020
473	0.11944	0.25	0.08827	0.989	0.028	0.85490	0.25	0.47911	1.020	0.022
523	0.11964	0.25	0.08777	0.997	0.030	0.85510	0.25	0.47861	1.006	0.023
573	0.12044	0.25	0.07309	0.999	0.023	0.85590	0.25	0.46393	0.982	0.016
623	0.12075	0.25	0.06778	0.994	0.022	0.85621	0.25	0.45862	0.991	0.015

673	0.11951	0.25	0.06604	1.026	0.018	0.85498	0.25	0.45688	1.013	0.011
723	0.12079	0.25	0.06120	0.980	0.018	0.85625	0.25	0.45200	1.031	0.011

Table S4 The detailed refinement results of SnSe-0.75%Te-0.7%Mo crystal with the temperature from 300 K to 723 K.

T (K)	Sn					Se				
	x	y	z	Occ	Uiso	x	y	z	Occ	Uiso
300	0.11990	0.25	0.09221	1.024	0.030	0.85537	0.25	0.48305	1.058	0.024
323	0.12046	0.25	0.08389	1.022	0.025	0.85593	0.25	0.47473	1.046	0.018
373	0.11990	0.25	0.08649	1.025	0.027	0.85537	0.25	0.47733	1.046	0.027
423	0.11941	0.25	0.08932	1.024	0.029	0.85488	0.25	0.48016	1.058	0.022
473	0.11957	0.25	0.08833	1.020	0.027	0.85504	0.25	0.47917	1.060	0.020
523	0.12039	0.25	0.08087	1.026	0.023	0.85586	0.25	0.47171	1.032	0.017
573	0.12032	0.25	0.07730	1.028	0.023	0.85579	0.25	0.46814	1.022	0.016
623	0.11951	0.25	0.06604	1.026	0.018	0.85498	0.25	0.45688	1.022	0.011
673	0.12072	0.25	0.06230	1.003	0.019	0.85618	0.25	0.45310	0.987	0.012
723	0.12009	0.25	0.06250	0.973	0.021	0.8555	0.25	0.45330	1.062	0.014

Detailed Comment 2: From all the PXRD refinements (synchrotron or not), and for each composition and *T*, detailed structural parameters must be provided: Basis, occupancies, reliability factors, angles distances etc....

Response: Thanks for your good suggestion. The detailed structural information of the samples has been added to the Supplementary Information.

Revision: (In the manuscript) The detailed structural data of the three samples is also provided in Tab. S2-S4.

(In the Supplementary Information)

Table S2 The detailed refinement results of SnSe crystal with the temperature from 300 K to 723 K.

T (K)	Sn					Se				
	x	y	z	Occ	Uiso	x	y	z	Occ	Uiso
300	0.12059	0.25	0.10008	1.015	0.03	0.85606	0.25	0.49092	0.953	0.023

323	0.12054	0.25	0.09999	0.997	0.028	0.85601	0.25	0.49083	1.008	0.022
373	0.11989	0.25	0.09714	1.000	0.029	0.85536	0.25	0.48799	1.004	0.022
423	0.11922	0.25	0.09575	1.004	0.025	0.85469	0.25	0.48660	1.007	0.018
473	0.11982	0.25	0.09403	1.003	0.026	0.85529	0.25	0.48488	0.996	0.020
523	0.11879	0.25	0.08438	0.990	0.023	0.85426	0.25	0.47523	1.019	0.016
573	0.12082	0.25	0.08031	0.996	0.027	0.85629	0.25	0.47116	1.025	0.020
623	0.12026	0.25	0.07596	0.997	0.025	0.85573	0.25	0.46681	0.999	0.019
673	0.11963	0.25	0.06570	1.006	0.020	0.85509	0.25	0.45650	0.974	0.013
723	0.11968	0.25	0.06456	1.007	0.018	0.85514	0.25	0.45660	0.972	0.012

Table S3 The detailed refinement results of SnSe-0.75%Te crystal with the temperature from 300 K to 723 K.

T (K)	Sn					Se				
	x	y	z	Occ	Uiso	x	y	z	Occ	Uiso
300	0.11985	0.25	0.09425	0.994	0.026	0.85531	0.25	0.48509	1.011	0.020
323	0.11982	0.25	0.09523	0.995	0.028	0.85524	0.25	0.48607	1.010	0.021
373	0.11970	0.25	0.09642	1.009	0.030	0.85516	0.25	0.48726	1.007	0.023
423	0.11957	0.25	0.09230	0.999	0.027	0.85504	0.25	0.48314	1.016	0.020
473	0.11944	0.25	0.08827	0.989	0.028	0.85490	0.25	0.47911	1.020	0.022
523	0.11964	0.25	0.08777	0.997	0.030	0.85510	0.25	0.47861	1.006	0.023
573	0.12044	0.25	0.07309	0.999	0.023	0.85590	0.25	0.46393	0.982	0.016
623	0.12075	0.25	0.06778	0.994	0.022	0.85621	0.25	0.45862	0.991	0.015
673	0.11951	0.25	0.06604	1.026	0.018	0.85498	0.25	0.45688	1.013	0.011
723	0.12079	0.25	0.06120	0.980	0.018	0.85625	0.25	0.45200	1.031	0.011

Table S4 The detailed refinement results of SnSe-0.75%Te-0.7%Mo crystal with the temperature from 300 K to 723 K.

T (K)	Sn					Se				
	x	y	z	Occ	Uiso	x	y	z	Occ	Uiso
300	0.11990	0.25	0.09221	1.024	0.030	0.85537	0.25	0.48305	1.058	0.024
323	0.12046	0.25	0.08389	1.022	0.025	0.85593	0.25	0.47473	1.046	0.018

373	0.11990	0.25	0.08649	1.025	0.027	0.85537	0.25	0.47733	1.046	0.027
423	0.11941	0.25	0.08932	1.024	0.029	0.85488	0.25	0.48016	1.058	0.022
473	0.11957	0.25	0.08833	1.020	0.027	0.85504	0.25	0.47917	1.060	0.020
523	0.12039	0.25	0.08087	1.026	0.023	0.85586	0.25	0.47171	1.032	0.017
573	0.12032	0.25	0.07730	1.028	0.023	0.85579	0.25	0.46814	1.022	0.016
623	0.11951	0.25	0.06604	1.026	0.018	0.85498	0.25	0.45688	1.022	0.011
673	0.12072	0.25	0.06230	1.003	0.019	0.85618	0.25	0.45310	0.987	0.012
723	0.12009	0.25	0.06250	0.973	0.021	0.8555	0.25	0.45330	1.062	0.014

Detailed Comment 3: structural data from all the compositions are not provided. The changes of cell parameters are only given for three compositions. The authors must concentrate their work on RT data and to determine if there is a linear change of cell parameters, bond angles versus x and y . It is very surprising that this has not been performed and presented in the present article. Long acquisition monochromated powder XRD data must be recorded for each composition.

Response: Thanks for your good suggestion. We have calculated the cell parameters and added them to Fig. S2. Results show that the cell parameters change linearly with the addition of Te and Mo, indicating that foreign elements are effectively alloyed into the matrix.

Revision: (In the manuscript) X-ray diffraction (XRD) was conducted to identify the phase of samples., which were found to exhibit a single phase of $Pnma$, as presented in Fig. S2, and the lattice parameters exhibit a near-linear variation. The EDS Mapping was also conducted to confirm that the element distribution is uniform in samples as shown in Fig S6. Results of XRD and EDS mapping collectively confirm that Te and Mo have been successfully alloyed into the matrix.

This variation trend is also consistent with the data in Fig. S2c and S2d.

(In the Supplementary Information)

Fig. S2 X-ray diffraction patterns on the cleavage plane. a SnSe-xTe. **b** SnSe-0.75%Te-yMo. **c** and **d** are the calculated lattice parameters of SnSe-xTe and SnSe-0.75%Te-yMo, respectively.

Detailed Comment 4: EDS mapping must be performed on each composition to attest the purity of the samples.

Response: Thanks for your good suggestion. The EDS Mapping has been carried out to show the distribution of elements and added to the Supplementary Information.

Revision: (In the manuscript) The EDS Mapping was also conducted to confirm that the element distribution is uniform in samples as shown in Fig S6. Results of XRD and EDS mapping collectively confirm that Te and Mo have been successfully alloyed into the matrix.

(In the Supplementary Information)

Fig. S6 Low-magnification scanning electron microscope (SEM) images and energy dispersive

spectrometer (EDS) mapping images of different elements. a1 SEM image of SnSe and the corresponding elemental mappings for **a2** Sn, **a3** Se, and **a4** Br. **b1** SEM image of SnSe-0.75%Te and the corresponding elemental mappings for **b2** Sn, **b3** Se, **b4** Br, and **b5** Te. **c1** SEM image of SnSe-0.75%Te-0.7%Mo and the corresponding elemental mappings for **c2** Sn, **c3** Se, **c4** Br, **c5** Te, and **c6** Mo.

After this in-depth structural analysis is performed, then the article can be resubmitted.

As a general comment, I don't think that the term of high symmetry is appropriate. Even the cell parameters are changed, the symmetry itself is not changed, as the space group is unchanged. there is no specific disorder induced by the anionic or cationic substitution which modify the symmetry. Then we can wonder if the change of properties is really affected by the change of symmetry. The lattice constrains and the final stoichiometry should be also considered

Response: Thanks for your good suggestion. The symmetry modification we talking about refers to the microscopic change in the crystal lattice, specifically, the changes in bond lengths and angles. As you mentioned, these changes occur without altering the space group (*Pnma*), which means the microscopic level modification within the *Pnma* phase. However, we have studied the continuous phase transition in SnSe from the low-symmetric *Pnma* to the high-symmetric *Cmcm* for several years. The characteristic angle (θ in Fig. 4f) can reflect the gradual transition process of symmetry and can establish a close relationship with the change in thermoelectric performance (refer to our previous work: the doi of 10.1002/aenm.201901334, 10.1126/science.abn8997 and 10.1038/s41467-023-37114-7). That is, the symmetry we mentioned can be regarded as a supplement to the symmetry of the space group. It is independent of the number of symmetry operations but does reflect microscopic changes in the crystal structure, which can strongly affect the thermoelectric transport. To further expound on the relationship between the subtle structural changes and the thermoelectric performance, we observed its microstructure and discussed it in the manuscript.

Revision:

To further confirm the symmetry enhancement of SnSe, we observed the microstructure of SnSe-0.75%Te-0.7%Mo. A detailed aberration-corrected scanning transmission electron microscopy (AC-STEM) characterization is summarized in **Fig. 5**. For comparison, an annular dark field- (ADF-) STEM image of pristine SnSe viewed along [010] direction is given in **Fig. 5a**, showing a distorted rocksalt structure. Some regions with darker contrasts are most likely due to intrinsic Sn vacancies. After Te/Mo co-alloying, an annular bright field- (ABF-) STEM image of such sample highlights the regions with darker contrast induced by strain (**Fig. 5b**). Enlarged ADF-STEM images of a representative strained region (**Fig. 5c** and **5d**) expose a distorted region with darker contrast within the lattice. An intensity profile across **Fig. 5d** shows a decrease in ADF-STEM image intensity at the Sn sites within the distorted region (**Fig. 5e**). Since Sn

vacancy will not induce such lattice distortion, the decreased intensity is most likely due to Mo substitution at Sn sites. We selected a highly distorted region in **Fig. 5c** and compared it with a reference region in **Fig. 5a**, the difference in lattice symmetry is obvious, which results in a discernible difference in θ as shown in **Fig. 5f1** and **5f2**. For the reference region, $\theta \sim 8.63^\circ$, close to the value in **Fig. 4f**, and for the distorted region, $\theta \sim 2.30^\circ$, which is significantly decreased and indicates the symmetry enhancement. To note, the distorted θ here is not the same as that in **Fig. 4f**, because the θ here is observed in the specific region while the θ obtained by refinement reflects the average value in the lattice. However, the decreased θ as a sign of the high symmetry can be confirmed by the refinement and microstructure observation.

Using inverse fast Fourier transform (IFFT) of selected reflections from **Fig. 5d**, we obtained IFFT images reconstructed using signals from only (020) planes (**Fig. 5g1**) and (004) planes (**Fig. 5g2**), and only (020) planes shows perturbation around the distorted region. The decrease in θ is achieved by the contraction between the farthest-neighboring Se-Sn pairs perpendicular to (020) planes (indicated by double arrows in **Fig. 5f1** and **5f2**), and such contraction is compensated by the insertion of extra planes parallel to (020) planes around the highly distorted region (**Fig. 5g**). Geometric phase analysis (GPA) further shows the compressive nature at the extra plane insertion position (**Fig. 5h1** and **5h2**). Moreover, the presence of strain fields revealed by GPA also indicates increased phonons scattering events caused by Mo substitution and lattice distortion.

Fig. 5 AC-STEM characterization of SnSe before and after Te/Mo co-doping. a ADF-STEM image of pristine SnSe viewed along [010] direction, inset showing the corresponding FFT spectrum. **b** ABF-STEM image of doped SnSe with an array of darker features induced by strain. **c** ADF-STEM image of a representative strained region, and **d** enlarged ADF-STEM image showing a distorted region in the lattice. **e** Intensity profile from boxed region in panel **d** (from left to right). **f1, f2** Boxed region indicated in panels **a** and **c**, respectively, showing enhancement in symmetry and decrease in θ after Te/Mo co-doping, overlaid by Sn and Se atom models. Double arrows

indicate the distance between the farthest-neighboring Se-Sn pair. **g1, g2** IFFT image using only (002) reflections and (400) reflections from panel **d**, respectively. **h1, h2** GPA of panel **d**.

One more comment: it is not specified in which direction of the samples, the electrical and thermal properties are measured. Is the texture similar for all the samples? this must be presented and discussed as the texture can strongly influence the transport properties. from only the XRD data on cleavage surfaces, it is impossible to determine the strength of the texture.

Response: Thanks for your good suggestion. All the samples are single crystals in this work, which are obtained by the Bridgman method. The single crystal structure can be confirmed by the diffraction patterns (has been added to the inset of Fig 5a) and XRD patterns. Considering the layered structure of SnSe, all samples can be divided into two directions: in-plane and out-of-plane. In this work, we focus on the in-plane thermoelectric performance as shown in Fig. S1, and ensure that measurement of electrical and thermal properties are in the same direction.

Revision: (In the manuscript) The photograph of the actual object of the obtained crystal and a schematic illustrating the in-plane direction are presented in Fig S1.

(In the Supplementary Information)

Fig. S1 The photo and schematic diagram of the SnSe crystal. a The photo of the SnSe crystal cleaved along (100) plane. b The schematic diagram of the SnSe crystal which shows how to cut out samples for thermoelectric transport measurement.

Referee 3:

General Comment: This study presents the critical role of Te and Mo alloying induced crystal symmetry modifications in SnSe. Improved symmetry results in converging conduction bands with increased carrier mobility ($\sim 422 \text{ cm}^2 \text{ V}^{-1} \text{ s}^{-1}$). Lattice thermal

conductivity is reduced to $\sim 1.1 \text{ W m}^{-1} \text{ K}^{-1}$ due to soft acoustic and optical phonon branches, with $zT \sim 0.6$ at 300 K, with an average ZT of ~ 0.89 over the range 300 – 723 K. A single-leg thermoelectric device based on this material achieves $\sim 5.3\%$ efficiency for a temperature gradient (ΔT) of ~ 300 K. Few inconsistencies as enumerated below requires a moderate revision prior to publication.

Detailed Comment 1: Clarify the dopability and solubility limits of Te and Mo in SnSe.

Response: Thanks for your good suggestion. The solubility limit of Te in SnSe is $\sim 1.5\%$ according to the existing report (doi: 10.1016/j.jallcom.2015.04.049), and the maximum alloying amount is 1.25% in our work, which is under the solution limit. On the other hand, there has been no research on the solubility limit of Mo in SnSe. However, there is no second phase observed in XRD patterns, and Mo is uniformly distributed in the SnSe matrix from the EDS Mapping. So we can confirm that Mo has successfully alloyed into SnSe and does not exceed the solution limit, as well as Te. Regarding this issue, we have emphasized it in the manuscript.

Revision: (In the manuscript) X-ray diffraction (XRD) was conducted to identify the phase of samples., which were found to exhibit a single phase of *Pnma*, as presented in Fig. S2, and the lattice parameters exhibit a near-linear variation. The EDS Mapping was also conducted to confirm that the element distribution is uniform in samples as shown in Fig S6. Results of XRD and EDS mapping collectively confirm that Te and Mo have been successfully alloyed into the matrix.

(In the Supplementary Information)

Fig. S2 X-ray diffraction patterns on the cleavage plane. a SnSe-xTe. **b** SnSe-0.75%Te-yMo. **c** and **d** are the calculated lattice parameters of SnSe-xTe and SnSe-0.75%Te-yMo, respectively.

Fig. S6 Low-magnification scanning electron microscope (SEM) images and energy dispersive spectrometer (EDS) mapping images of different elements. a1 SEM image of SnSe and the corresponding elemental mappings for **a2** Sn, **a3** Se, and **a4** Br. **b1** SEM image of SnSe-0.75%Te and the corresponding elemental mappings for **b2** Sn, **b3** Se, **b4** Br, and **b5** Te. **c1** SEM image of SnSe-0.75%Te-0.7%Mo and the corresponding elemental mappings for **c2** Sn, **c3** Se, **c4** Br, **c5** Te, and **c6** Mo.

Detailed Comment 2: How do the lattice parameters of SnSe vary with increasing Mo content and decreasing Te content? In addition to discussing the optimal sample, the

results for other Mo- and Te-based samples can also be included.

Response: Thanks for your good suggestion. We have calculated the lattice parameters of all SnSe samples by the XRD results. Its variation trend is almost linear and consistent with the result of SR-XRD.

Revision: (In the manuscript) X-ray diffraction (XRD) was conducted to identify the phase of samples., which were found to exhibit a single phase of *Pnma*, as presented in Fig. S2, and the lattice parameters exhibit a near-linear variation.

This variation trend is also consistent with the data in Fig. S2c and S2d.

(In the Supplementary Information)

Fig. S2 X-ray diffraction patterns on the cleavage plane. a SnSe-xTe. **b** SnSe-0.75%Te-yMo. **c** and **d** are the calculated lattice parameters of SnSe-xTe and SnSe-0.75%Te-yMo, respectively.

Detailed Comment 3: The synthesized crystal appears to be polycrystalline and is not well characterized or described in terms of its crystallinity. The single-crystal SnSe demonstrates impressive ZT values ranging from 2.2 to 2.6 at 913 K. In contrast, polycrystalline versions of SnSe, synthesized through various methods, exhibit significantly lower overall ZT values, which can be compared to elucidate the relevance of crystal symmetry and grain boundary effects.

Response: The synthesized samples in this work are crystals. The XRD patterns in Fig.

S2 were carried out by scanning the cleavage plane and only (100) crystal plane was identified, indicating the characteristic of crystals. In addition, the ZT over 2 is obtained along the out-of-plane direction in n -type SnSe crystals. However, this work focuses on the in-plane thermoelectric performance, due to the better mechanical properties for device fabricating. As a layered material, SnSe performs different thermoelectric transports along different directions (in-plane and out-of-plane). So we cannot solely judge the samples as polycrystals based on their ZT values. To preclude any potential ambiguity, we have incorporated photographs of the crystal alongside schematic diagrams delineating the orientation of the testing direction.

Revision: (In the manuscript) The photograph of the actual object of the obtained crystal and a schematic illustrating the in-plane direction are presented in Fig S1.

(In the Supplementary Information)

Fig. S1 The photo and schematic diagram of the SnSe crystal. a The photo of the SnSe crystal cleaved along (100) plane. b The schematic diagram of the SnSe crystal which shows how to cut out samples for thermoelectric transport measurement.

Detailed Comment 4: The continuous phase transition of SnSe from a low-symmetric $Pnma$ to a high-symmetric $Cmcm$ phase between 600 K and 800 K may affect the long-term stability and reliability of SnSe-based thermoelectric devices. Justify the thermal stability of the synthesized samples.

Response: Thanks for your good suggestion. The thermal stability determines whether the material can continue working for a long time. Here we carried out three thermal cycling measurements of the best-optimized sample (SnSe-0.75%Te-0.7%Mo) to evaluate its thermal stability. Results showed good stability in thermoelectric performance, and we have added them into the Supplementary Information.

Revision: (In the manuscript) Furthermore, we conducted two cycles of heating and cooling tests on the sample as shown in Fig. S16, which demonstrated good thermal stability.

(In the Supplementary Information)

Fig. S16 The thermal stability of thermoelectric properties in SnSe-0.75%Te- μ Mo. a Electrical conductivity. **b** Seebeck coefficient. **c** Power factor. **d** Total thermal conductivity. **e** Lattice thermal conductivity **f** ZT value.

Detailed Comment 5: While the study reports improvements in n-type SnSe, the achieved power factor ($\sim 28 \mu\text{W cm}^{-1} \text{K}^{-2}$) and ZT (~ 0.6) at 300 K are still relatively low compared to state-of-the-art p-type SnSe materials, which have shown power factors up to $\sim 85 \mu\text{W cm}^{-1} \text{K}^{-2}$ and ZT of ~ 1.4 at ambient temperature. This disparity in performance between n-type and p-type SnSe needs to be emphasized more clearly in results and discussion.

Response: Thanks for your good suggestion. The comparison between n-type and p-

type SnSe is an interesting topic, and this can help us better understand the thermoelectric transport mechanism in SnSe and other material systems. As a prerequisite, the performance of *n*-type SnSe in this work and the high performance of *p*-type SnSe are both along the in-plane direction, so the difference caused by anisotropy can be ruled out. Doping efficiency is an important factor, where the carrier concentration can reach $\sim 4 \times 10^{19} \text{ cm}^{-3}$ in Na-doped *p*-type SnSe but only $\sim 1.2 \times 10^{19} \text{ cm}^{-3}$ in Br-doped *n*-type SnSe. Meanwhile, there are 6 valence bands and 3 or 4 of them usually participate in the thermoelectric transport in *p*-type SnSe. But in *n*-type SnSe, only 2 conduction bands can be utilized. Also, the synglisis which contains the momentum and energy alignment of energy bands was found in *p*-type SnSe, but it not appears in *n*-type SnSe. The detailed discussion has been added to the manuscript.

Revision: Thermoelectric devices require comparable performance between *n*-type and *p*-type materials. Therefore, to advance the development of the *n*-type SnSe crystal and match it with the high-performance *p*-type one, we compare the thermoelectric transports in *n*-type and *p*-type SnSe for guidance. First, the doping of Na in SnSe is more effective than that of halogen. The carrier concentration can reach $\sim 4 \times 10^{19} \text{ cm}^{-3}$ in Na-doped SnSe, which is approximately 4 to 5 times that of Br doping and much higher than Cl and I doping. Therefore, cations with high valence states are expected to be introduced to *n*-type SnSe. Some efforts have already been made, such as La, Ce, and W, but more effective elements are needed. Second, there are 6 valence bands in SnSe and 3 or 4 of them can participate in *p*-type electrical transport, while only 2 conduction bands can be utilized for *n*-type electrical transport. Meanwhile, the synglisis including the momentum and energy alignment of valence bands synergistically optimize the m^* and μ in *p*-type SnSe. In contrast, strategies of band sharpening and resonant level inducing can be conducted in *n*-type SnSe to achieve a similar effect as synglisis.

Detailed Comment 6: *The temperature-dependent behavior of the heat capacity and its role in influencing the thermal transport properties across the studied temperature range, needs to be established well.*

Response: Response: Thanks for your good suggestion. As an important parameter in thermal transport, the heat capacity (C_p) is indispensable. We obtained the C_p by the Debye model and have provided it in Supplementary Information.

Revision: (In the manuscript) The detail of the thermal transport is shown in Fig. S3d-S3h and S4d-S4h, including the κ_{tot} , κ_{lat} , and other thermal transport parameters of all samples, and the heat capacity is shown in Fig S5.

(In the Supplementary Information)

Fig. S5 The heat capacity (C_p) as the function of temperature. **a** SnSe- x Te and **b** SnSe-0.75%Te- y Mo.

Detailed Comment 7: What is the grain size? The lack of structural characterization raises concerns about elemental distribution and sample stoichiometry.

Response: Thanks for your good suggestion. As we discussed above, the samples in this work are single crystals, which have no grain boundaries. So it is not necessary to discuss the grain size. We carried out the EDS Mapping to observe the elemental distribution in samples, which has been added in Supplementary Information. Results show that the distribution of all elements is uniform.

Revision: (In the manuscript) The EDS Mapping was also conducted to confirm that the element distribution is uniform in samples as shown in Fig S6.

(In the Supplementary Information)

Fig. S6 Low-magnification scanning electron microscope (SEM) images and energy dispersive spectrometer (EDS) mapping images of different elements. **a1** SEM image of SnSe and the corresponding elemental mappings for **a2** Sn, **a3** Se, and **a4** Br. **b1** SEM image of SnSe-0.75%Te and the corresponding elemental mappings for **b2** Sn, **b3** Se, **b4** Br, and **b5** Te. **c1** SEM image of SnSe-0.75%Te-0.7%Mo and the corresponding elemental mappings for **c2** Sn, **c3** Se, **c4** Br, **c5** Te, **c6** Mo.

and c6 Mo.

Detailed Comment 8: There are prevailing typos and errors which needs to be thoroughly checked and corrected.

Response: Thanks for your kind reminder. We have noticed some typos and errors in the manuscript and corrected them.

Revision: GSAS software was used to analyze the test data and the accurate structural information of SnSe samples in this work was obtained.

Raw materials, including Sn (bulk, 5N), Se (shot, 4N), Te (bulk, 5N), Mo (powder, 3N), and SnBr₂ (powder, 2N), were used to synthesize the series of SnSe_{0.97-x}Te_xBr_{0.03} ($x = 0, 0.5\%, 0.75\%, 1\%, 1.25\%$) and Sn_{1-y}Mo_ySe_{0.9625}Te_{0.0075}Br_{0.03} ($y = 0, 0.3\%, 0.5\%, 0.7\%, 0.9\%$) crystal samples.

We hope that the revised manuscript is now suitable for publication in *Nature Communications*.

Sincerely and best regards

Li-Dong Zhao

Jan. 23, 2025

Referee 1:

General Comment: Authors discussed answers nicely, I recommend publishing in *Nature Comm.*

Response: Thanks for your affirmation of our work, and the suggestions you provided have made our manuscript more complete.

Referee 2:

General Comment: The authors clearly addressed all of my comments.

Response: Thanks for your good suggestions on the characterization of crystal structure. We are honored that our response can be recognized.

Referee 3:

General Comment: The authors have addressed the questions and considerations, enhancing the clarity of their research. Therefore, the manuscript is recommended for acceptance.

Response: Thanks for your instructive comments on our work and it is a pleasure to receive your approval.

The paper has addressed the editorial requests and we have provided the related files. We hope this work is now suitable for publication in *Nature Communications*.

Sincerely and best regards

Li-Dong Zhao

Feb 8, 2025